# Long ncRNA A-ROD activates its target gene DKK1 at its release from chromatin

Evgenia Ntini[1], Annita Louloupi[1,2], Julia Liz[1], Jose M. Muino[3], Annalisa Marsico[1,2] & Ulf Andersson Vang Ørom [1,4]

Long ncRNAs are often enriched in the nucleus and at chromatin, but whether their dissociation from chromatin is important for their role in transcription regulation is unclear. Here, we group long ncRNAs using epigenetic marks, expression and strength of chromosomal interactions; we find that long ncRNAs transcribed from loci engaged in strong long-range chromosomal interactions are less abundant at chromatin, suggesting the release from chromatin as a crucial functional aspect of long ncRNAs in transcription regulation of their target genes. To gain mechanistic insight into this, we functionally validate the long ncRNA A-ROD, which enhances DKK1 transcription via its nascent spliced released form. Our data provide evidence that the regulatory interaction requires dissociation of A-ROD from chromatin, with target specificity ensured within the pre-established chromosomal proximity. We propose that the post-transcriptional release of a subset of long ncRNAs from the chromatin-associated template plays an important role in their function as transcription regulators.

[1] Max Planck Institute for Molecular Genetics, 14195 Berlin, Germany. [2] Free University Berlin, 14195 Berlin, Germany. [3] Humboldt University, 10115 Berlin, Germany. [4] Institute for Molecular Biology and Genetics, Aarhus University, 8000 Aarhus, Denmark. Correspondence and requests for materials should be addressed to E.N. (email: ntini@molgen.mpg.de) or to U.A.V.Ø (email: ulf.orom@mbg.au.dk)

Thousands of long non-coding RNAs (ncRNAs) are encoded in the genome, fulfilling regulatory functions in development and disease. Long ncRNAs are distinct from short ncRNA species in that they are processed through the same molecular machinery as mRNAs and resemble mRNAs in their organization with several exons as well as a 5′ cap and frequent polyadenylation[1]. To exert function, long ncRNAs can interact with proteins and direct their binding to DNA[2,3] or enhance their enzymatic activity[4,5]. Long ncRNAs are often enriched in the nucleus and in some cases tethered to chromatin suggesting an involvement in epigenetic regulation[6,7]. Long ncRNAs are emerging as key players in transcription regulation, exerting both positive and negative activity[1]; through currently emerging sequence-structure motifs, some long ncRNAs have been shown to target promoters[8] or interact with chromatin[9], while others mediate transcriptional interference of antisense overlapping genes without leaving the site of transcription[10]. Yet, while the nuclear localization of most long ncRNAs seems intuitive for their role in transcription regulation, it is intriguing to what extent their activity depends on association to chromatin[11].

Enhancers, as distant regulatory elements involved in transcription regulation of target gene expression, can be identified via epigenetic marks, including histone marks and DNA methylation. Functional enhancers have been described as demarcated transcription units with relatively elevated levels of the histone 3 lysine 4 trimethylation (H3K4me3) histone mark[12,13] giving rise to long, spliced, and polyadenylated ncRNAs which in turn mediate the enhancer function and transcription activation of adjacent target genes[2,4,14–19]. Enhancers can also be pervasively transcribed into relatively short and unstable enhancer-associated ncRNAs called eRNAs[20–25]. These two groups, activating long ncRNAs and eRNAs, are distinct in terms of production, nature and stability, although our understanding of the functional repertoire of ncRNAs transcribed from active enhancers is not yet complete[1,16,26,27].

Here, we find that, among long ncRNAs actively expressed in MCF-7 breast-cancer cells, those transcribed from loci engaged in strong RNA polymerase II (Pol II)-dependent chromosomal interactions to target gene promoters are less enriched in the chromatin-associated RNA fraction. Through functional analyses of the long ncRNA A-ROD (Activating Regulator of DKK1), we show that transcriptional enhancement of its target gene DKK1 is accompanied by an A-ROD-dependent recruitment of the general transcription factor EBP1 to the DKK1 promoter. Detailed validation provides evidence that the regulatory effect is exerted by A-ROD at its release from the chromatin-associated site of transcription. The mechanistic insights gained from this study establish that dissociation of activating long ncRNAs from chromatin is necessary to mediate RNA-dependent regulation of their target gene expression, adding an important new mechanistic perspective to the functional repertoire of long ncRNAs.

## Results

### Grouping of long ncRNAs by expression and chromatin features.

Intra-chromosomal and inter-chromosomal chromatin interactions are important for precise regulation of gene expression[28,29]. To determine the characteristics of long ncRNAs engaged in long-range chromatin interactions we used Pol II-dependent ChIA-PET data (Chromatin Interaction Analysis by Paired-End Tag sequencing) from MCF-7 cells[29,30]. ChIA-PET employs a chromatin immunoprecipitation (ChIP) step followed by high-throughput sequencing to detect long-range interactions at regions bound by a target protein of interest, in this case Pol II. Enriching for loci bound by a specific protein increases the probability of detecting regulatory interactions. This technique

has been used to map enhancer-promoter interactions[31]. To define a working dataset, we combined the ENCODE annotation of long ncRNAs, long ncRNAs reported in ref. [32] and de novo transcript assembly in MCF-7 cells from chromatin-associated RNA-sequencing data (CHR-RNA-seq) (Methods: Data Availability). To facilitate the analysis of chromatin marks at specific loci, without confounding signals from overlapping genes, and in order to obtain high-confidence long ncRNAs, we required transcripts to have at least one splicing junction, no coding potential (using CPAT[33]) and no overlap with annotated protein-coding genes. We obtain a list of 12,553 long ncRNAs transcribed from 10,606 genomic loci. This set was further narrowed down to 4467 long ncRNAs with detectable expression in MCF-7 cells, used in all subsequent analyses (Fig. 1a, Supplementary Data 1, and Methods section).

We grouped the long ncRNAs using histone marks associated with transcription regulation, namely H3K4me1[34], H3K27Ac and H3K4me3[30]; RNA expression including GRO-seq[35] and our CHR-RNA-seq; DNA methylation; and Pol II ChIA-PET interaction data from MCF-7 cells[29,30]. Using ENCODE DNA methylation data we found a correlation between promoter DNA hypomethylation and long ncRNA expression specificity across nine cell lines, suggesting an involvement of the DNA methylation status in long ncRNA expression (Supplementary Fig. 1a). We extended the dataset of long ncRNA promoters with assayed DNA methylation in MCF-7 cells by array-based DNA bisulfite sequencing (Methods; Supplementary Fig. 1b). We then performed k-means clustering of 2685 long ncRNA transcripts according to epigenetic marks and transcription, where all data are available, grouping into 15 clusters (Supplementary Fig. 1c). This shows that low levels of DNA methylation coincide with high expression of long ncRNAs, association of enhancer marks and engagement in strong ChIA-PET interactions. We thereby excluded DNA methylation from the analysis to prevent limitations in the number of the examined loci and performed k-means clustering for the 4467 long ncRNAs of the initial dataset. Clusters with strong ChIA-PET interactions are marked with promoter (high H3K4me3) or enhancer (high H3K4me1 and/or H3K27ac) histone marks (Fig. 1b). Cluster 6 is of particular interest, since it contains 104 long ncRNAs showing enrichment of all included histone marks, relatively high expression and engagement in strong chromosomal interactions, high-scored by ChIA-PET. As expected, the promoters of the long ncRNAs of cluster 6 with assayed DNA methylation are commonly hypomethylated (Supplementary Fig. 1d). In addition, the intermediate H3K4me1/H3K4me3 ratio of cluster 6 is characteristic of actively transcribed enhancers[26] (Fig. 1c). The positional profile of histone marks shows characteristic bidirectional distribution around transcription start sites (TSS), with cluster 6 showing on average the highest levels of H3K27Ac, a hallmark of active enhancers (Supplementary Fig. 1e-g).

### Differential chromatin-association of long ncRNAs.

Long ncRNAs are more abundant in the chromatin fraction, when compared to mRNAs[6,36], and this is thought to reflect their nuclear functions. We assessed chromatin-association of long ncRNAs in MCF-7 cells by normalizing CHR-RNA-seq to ENCODE available nuclear polyA+ RNA-seq (log2 RPKM ratio). Interestingly, long ncRNAs of clusters with higher ChIA-PET scores are significantly less enriched in the chromatin fraction (Fig. 1d, e). This suggests that dissociation from chromatin could play a functional role for long ncRNAs transcribed from loci engaged in strong long-range chromosomal interactions. Plotting the DESeq2[37] generated log2 fold-changes of expression generates a similar result (Supplementary Fig. 2a). To exclude any bias in

the analysis using nuclear polyA+ RNA-seq, we sequenced total nucleoplasmic RNA (after depletion of ribosomal species) and re-assessed chromatin-association as the ratio of total chromatin-associated to nucleoplasmic RNA. This results in a similar overall distribution, with clusters 4, 6, and 13 of strong chromosomal interactions showing on average lower chromatin association (Supplementary Fig. 2b). Because of the observed difference in the

distribution by using nuclear polyA+ data, we conducted further analysis to examine what defines the chromatin-association of long ncRNAs. We performed linear regression with the parameters GRO-seq, H3K4me3, H3K4me1, ChIA-PET, and H3K27Ac (Supplementary Fig. 2c). This results in the ChIA-PET interaction strength and the H3K4me3 mark producing the highest negative coefficients in predicting chromatin-association

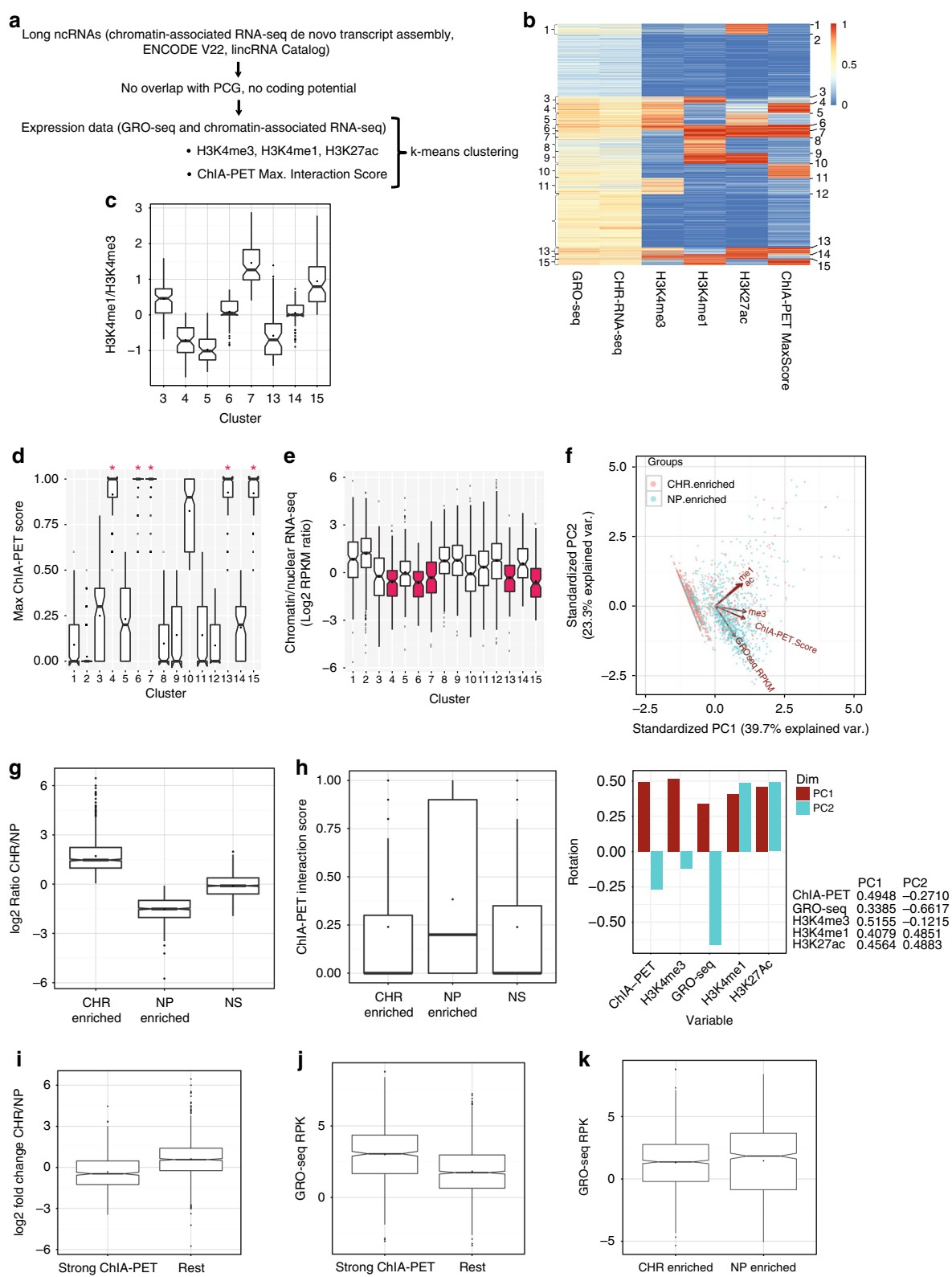

with a fairly good linear fit (correlation = 0.476). The negative sign of the coefficients means a significant negative contribution, i.e., strong ChIA-PET interactions and high H3K4me3 define low chromatin association. Including nuclear polyA+ expression as an extra parameter in the linear regression fit improves the correlation only slightly (correlation = 0.487), with ChIA-PET interaction strength and H3K4me3 remaining the most significant parameters in determining the fit (Supplementary Fig. 2d). We also employed principal component analysis (PCA), based on the same features used for linear regression, to explore patterns in our data and compare chromatin- vs. nucleoplasmic-enriched long ncRNAs, (Fig. 1f). Briefly, points are projected in a new two-dimensional variable space, defined by principal components PC1 and PC2, where PC1 and PC2 are linear combinations of the original five variables that maximize and explain most of the variability in our data (see also Methods section). The PCA result shows that nucleoplasmic-enriched long ncRNAs (blue points) form a distinct cluster separate from chromatin-enriched long ncRNAs (red points). The bigger positive contribution to the first component PC1 is given by the ChIA-PET interaction strength and H3K4me3 (Fig. 1f, barplot). This is in agreement with the result of the linear regression analysis. We then performed differential expression analysis using DESeq2[37] to extract significantly chromatin- vs. nucleoplasmic-enriched long ncRNAs (at $p$-adjusted < 0.1; Supplementary Data 2 and 3). Importantly, nucleoplasmic-enriched long ncRNAs engage in strong ChIA-PET interactions with significantly higher scores compared to chromatin-enriched long ncRNAs or to long ncRNAs not enriched in either of the two fractions (Fig. 1g, h). In agreement, long ncRNAs deploying strong interactions are on average significantly less enriched in the chromatin fraction (Fig. 1i). Using GRO-seq data[35] as a direct measurement of transcription, we observe that long ncRNAs of strong chromosomal interactions show higher average expression (Fig. 1j). This is expected since the ENCODE ChIA-PET data maps long-range interactions associated with Pol II. Importantly however, there is no significant difference in the expression range between significantly nucleoplasmic-enriched and chromatin-enriched long ncRNAs (using DESeq2 log2 fold-changes at $p$-adjusted < 0.1; Fig. 1k). Even by setting a more stringent cutoff and comparing between the strongest chromatin-enriched (fold-change CHR/NP > 2) and strongest nucleoplasmic-enriched (fold-change CHR/

NP < 0.5) entries there is no bias (Supplementary Fig. 2e). There is also no direct correlation between expression and the degree of chromatin-association (Supplementary Fig. 2f). Therefore the transcription activity of long ncRNAs does not per se seem to determine the degree of chromatin-association. Together these data indicate that long ncRNAs transcribed from loci engaged in strong long-range chromatin interactions show significantly lower chromatin-association. This implies that the release of these long ncRNAs from chromatin is important for their function.

**Long ncRNA-target gene pairs show coordinated expression**. The MCF-7 cell line is an estrogen receptor alpha (ERa)-positive breast cancer cell line and estradiol (E2) treatment induces broad transcriptional changes. To examine the involvement of long ncRNAs in transcription regulation of target genes predicted by ChIA-PET, under different conditions, we used expression data from control (0 min) and 40 min E2-treated MCF-7 cells, in particular GRO-seq[35] was used as a direct measurement of transcription. The E2-mediated transcriptional changes of the long ncRNAs and ChIA-PET predicted target genes correlate well (Supplementary Fig. 3a) and the transcriptional response of long ncRNAs to E2 generally precedes that of their interacting genes (Fig. 2a, b, Supplementary Fig. 3b). This is in agreement with previous studies reporting that transcriptional responses of stimuli-activated enhancers temporally precede expression changes of target genes[38–42]. The lag in the time response of the target genes compared to long ncRNAs is greater for the E2-downregulated (Fig. 2a) vs. E2-upregulated (Fig. 2b) long ncRNA-target gene pairs, and absent from random gene–gene pairs (Supplementary Fig. 3b). This difference may be due to upregulated genes being significantly enriched in ERa-binding sites[43], which correlates with an increase in expression triggered as a direct response to ERa binding (Supplementary Fig. 3c). The above data are in agreement with long ncRNAs being involved in positive regulation of their Pol II ChIA-PET identified interacting target genes.

**A-ROD as a tissue-specific regulator of DKK1 expression**. We further examined cluster 6 due to the prominent presence of enhancer marks and high chromatin-dissociation. We ranked the long ncRNA-interacting gene pairs of cluster 6 according to

---

**Fig. 1** Clustering and chromatin-association of long ncRNAs. **a** Schematic representation of the pipeline followed to define the final set of long ncRNAs used in this study, and the k-means clustering parameters. **b** K-means clustering of 4467 long ncRNAs. Clustering parameters are histone mark signal values (ChIP-seq peaks from H3K4me3, H3K4me1, and H3K27ac in MCF-7), expression (GRO-seq[35] and chromatin-associated RNA-seq log(RPKM)), and ChIA-PET maximum interaction scores. All values were rescaled in the range 0 to 1 using min–max normalization. **c** H3K4me1/H3K4me3 ratio for relevant clusters, $y$-axis is in log scale. **d** Comparison of ChIA-PET interaction scores for all clusters. Marked are the clusters with the strongest interactions (ChIA-PET score median = 1 and mean > 0.9) (clusters 4, 6, 7, 13, 15). **e** Comparison of the long ncRNA chromatin-association character for all clusters. The ratio chromatin-associated RNA-seq RPKM to nuclear polyA+ RPKM is plotted in $\log_2$ scale. Clusters with significantly lower chromatin-association are red-marked (Wilcoxon-Mann-Whitney test $p$-value < 2.2e-16). **f** Principal component analysis (PCA) biplot showing the multivariate variation of chromatin-enriched (log2 chromatin-association > 0, red) and nucleoplasmic-enriched (log2 chromatin-association < 0, blue) long ncRNAs in terms of five variables: ChIA-PET, GRO-seq, H3K4me3, H3K4me1, H3K27Ac. Points represent projections of the original values in a new space, defined by a new set of variables, the principal components PC1 and PC2, which better explain the patterns in the data. The contribution of the initial variables in explaining the variance within each PC is shown in the barplot underneath. **g** Boxplot displaying the distribution of the *DESeq2* derived log2 fold changes in expression using total chromatin-associated vs. nucleoplasmic RNA-seq, for 1154 significantly chromatin-enriched long ncRNAs (at *DESeq2* $p$-adjusted < 0.1; Supplementary Data 2), 968 significantly nucleoplasmic-enriched long ncRNAs (Supplementary Data 3) and 1807 long ncRNAs not enriched in either of the two fractions (non-significant, 'NS'). **h** Boxplot displaying the distribution of ChIA-PET interaction scores for the three sets of long ncRNA defined in (**g**). Nucleoplasmic-enriched long ncRNAs show on average significantly higher ChIA-PET interaction scores (Wilcoxon-Mann-Whitney $p$-value = 9.462e-16 to CHR-enriched and $p$-value < 2.2e-16 to NS). **i** Boxplot displaying the distribution of *DESeq2* generated log2 fold changes CHR/NP for long ncRNAs engaged in strong ChIA-PET interactions (clusters 4, 6, 7, 13, 15; $n = 664$) and long ncRNAs not engaged in ChIA-PET ('rest'; clusters 3, 5, 12, 14; $n = 1402$). Wilcoxon-Mann-Whitney $p$-value < 2.2e-16. **j** Boxplot displaying the distribution of expression (GRO-seq[35] reads per kb, RPK) of long ncRNAs either engaged or not in strong ChIA-PET interactions, defined as in (**i**). Wilcoxon-Mann-Whitney test $p$-value < 2.2e-16. **k** Boxplot displaying the distribution of expression (GRO-seq RPK) of the significantly chromatin enriched ('CHR enriched', $n = 1154$) and nucleoplasmic enriched ('NP enriched', $n = 968$) long ncRNAs

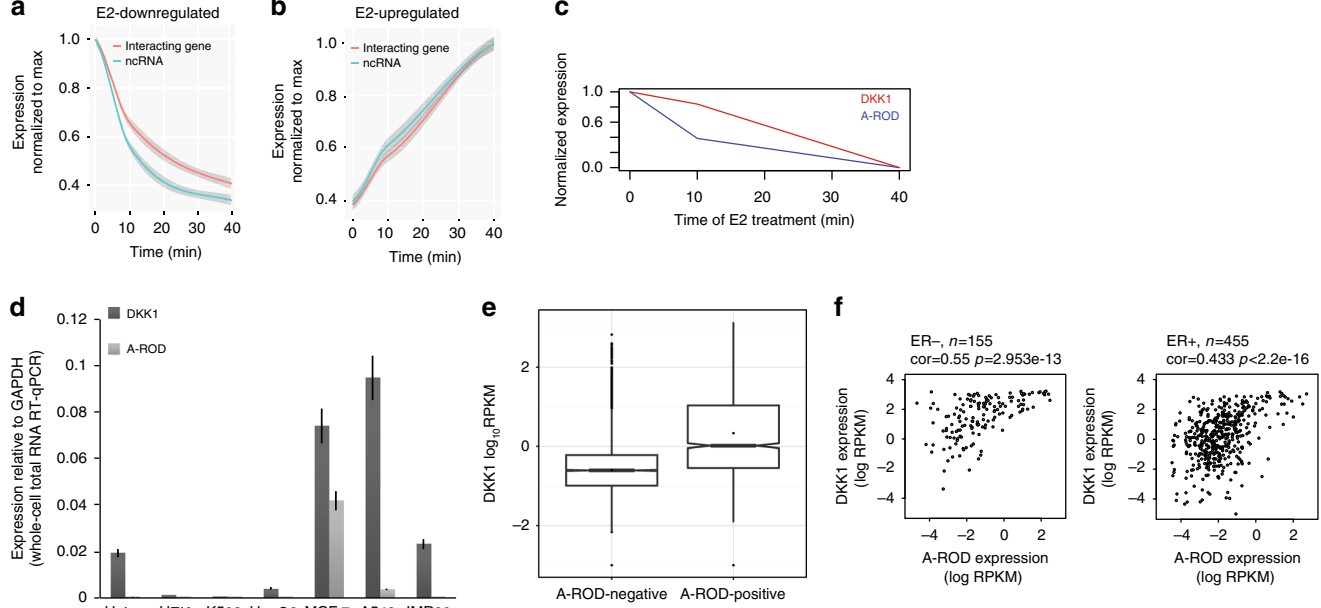

**Fig. 2** Expression of interacting long ncRNA-target gene pairs. **a** E2 mediated response for downregulated long ncRNA-interacting gene pairs ($n = 419$ with ChIA-PET interaction score > 200). Expression is normalized to maximum, i.e., GRO-seq[35] RPKM from 10 and 40 min of E2 treatment are normalized to 0 h (EtOH control). Fitted regression model lines (loess curves) are drawn. The 95% confidence region is grey shaded. **b** Same as in (**a**), but for the E2-upregulated long ncRNA-gene pairs ($n = 512$, ChIA-PET interaction score > 200). Expression is normalized to maximum, i.e., to the 40 min E2-treatment GRO-seq RPKM. **c** A-ROD and DKK1 expression as a time response to E2 induction. GRO-seq RPKM from 0 h (EtOH control), 10 and 40 min of E2 treatment were rescaled in the range 0 to 1 using min–max normalization. **d** Expression of A-ROD and DKK1 measured by RT-qPCR across seven cell lines, normalized to GAPDH. **e** Expression of DKK1 in A-ROD–positive ($n = 1659$) and A-ROD–negative tissue-samples ($n = 6896$) using GTEx data. Wilcoxon-Mann-Whitney test $p$-value < 2.2e-16. **f** Correlation of A-ROD and DKK1 expression (total exon RPKM in log scale) in ER positive and ER negative breast cancer samples. Data obtained from the Cancer Genome Atlas

changes of expression upon 40 min E2-treatment vs. control in MCF-7 cells. For further functional validation we assessed downregulated interacting pairs without ERa binding sites determined by ChIP-seq[43] +/−20 kb from the TSS to avoid direct transcriptional responses to E2. The top candidate is the long ncRNA *A-ROD* transcribed 130 kb downstream of Dickkopf homolog 1 (DKK1) on the opposite strand (Supplementary Fig. 4a). The DKK1 protein is involved in regulating the Wnt signaling pathway in many tissues and in breast cancer[44–46]. E2-downregulation of A-ROD precedes that of DKK1 (Fig. 2c) in accordance with the generally faster transcriptional response of long ncRNAs (Fig. 2a). A-ROD is a relatively highly expressed, multi-exonic and polyadenylated long ncRNA (Supplementary Fig. 4a-b, Supplementary Fig. 5). There are two isoforms expressed in MCF-7, one major and one minor, arising from alternative splicing at the first exon–intron junction.This is supported by the ENCODE nuclear polyA+ RNA-seq data as well as de novo assembly from our chromatin-associated total RNA-seq (Supplementary Fig. 5a-b), and verified by semi-quantitative RT-PCR (Supplementary Fig. 5c). The above data denote A-ROD as a bona-fide long ncRNA transcribed from an active enhancer, rather than an enhancer-associated eRNA[24,25,27]. The expression of A-ROD in MCF-7 cells is high compared to other cell lines (Fig. 2d) and generally shows an expression restricted to epithelial and endothelial cells (NHEK and HMEC)[30]. Using data from the Genotype-Tissue Expression Project (GTEx), we observe a correlation between DKK1 and A-ROD expression among several tissue samples (Supplementary Fig. 4c). In addition, there is a strong and significant quantitative difference in DKK1 expression between A-ROD–positive and A-ROD–negative samples (Fig. 2e), and a significant correlation between DKK1 and A-ROD expression across a panel of both ERa-positive and ERa-negative

breast cancer samples (Fig. 2f). Taken together these data implicate A-ROD as a tissue-specific regulator of DKK1 expression, as was further studied.

**A-ROD enhances DKK1 transcription.** Knock-down of enhancer-like long ncRNAs using siRNAs dampens the expression of their target genes[2,4,15,18], but where exactly in the biogenesis process the RNAs are targeted and functional is unknown. Upon knock-down of A-ROD using different siRNA sequences in MCF-7 cells, there is a reduction in DKK1 mRNA levels comparable to the extent of A-ROD depletion (Fig. 3a, Supplementary Fig. 4d). SiRNA-mediated depletion of DKK1 does not affect A-ROD levels. Neither A-ROD nor DKK1 siRNA-mediated depletion affect the levels of two unrelated transcripts (*LINC-616M22* and *OLMALINC*). To further determine whether A-ROD regulates DKK1 at the transcriptional level, we performed ChIP for Pol II in control and siRNA treated cells, using the N-20 antibody (Fig. 3b). We see a significant decrease in total Pol II association at the DKK1 gene body, whereas there is no significant change in total Pol II occupancy at either GAPDH or A-ROD loci, supporting a specific regulatory effect of A-ROD on DKK1 transcription. These data are in agreement with a direct transcriptional effect on DKK1 mediated by the long ncRNA A-ROD.

**Chromosome conformation to target genes is pre-established.** Enhancers are looped to promoters of regulated genes in the genome through long-range chromosomal interactions[28]. The effect of long ncRNAs in determining the interaction landscape is not firmly established. Although some long ncRNAs may affect long-range chromosomal interactions[4,47], other studies report no

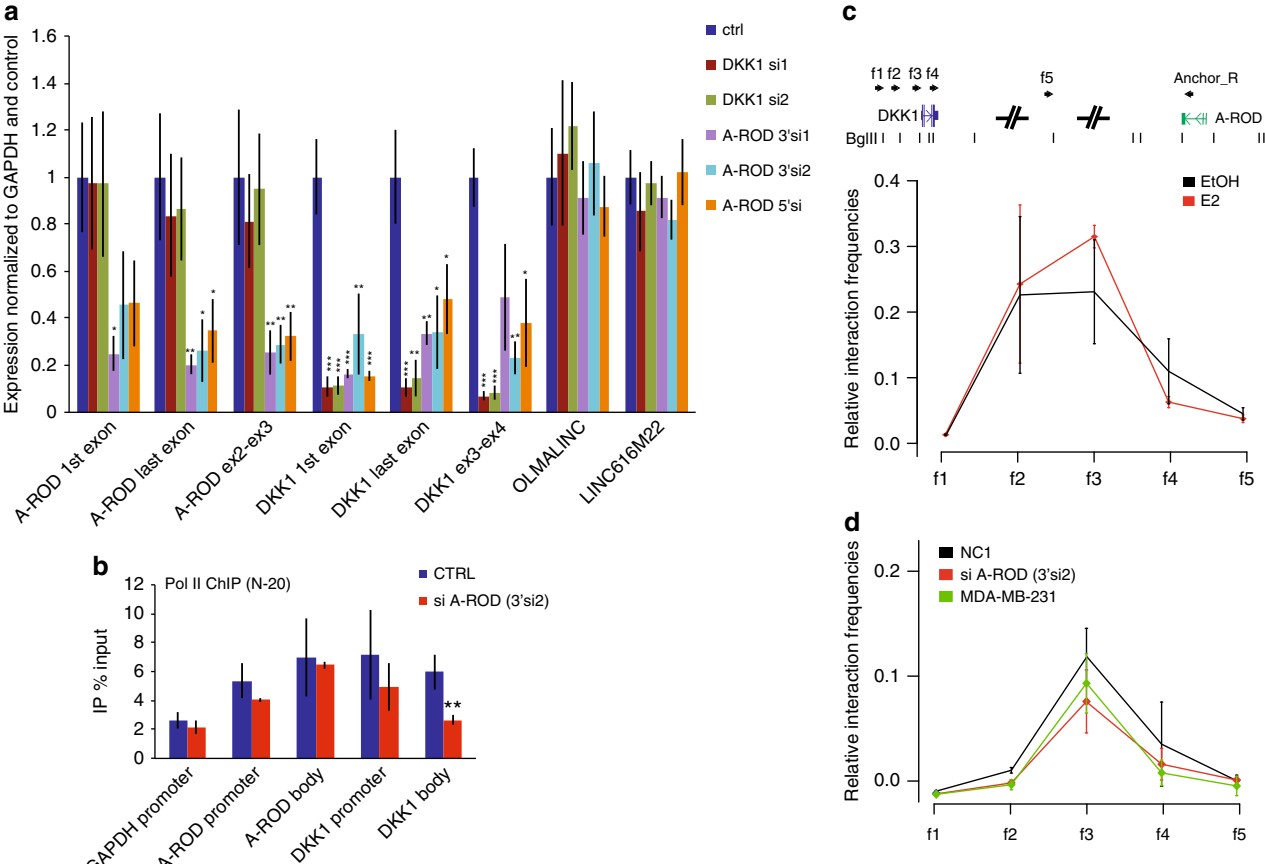

**Fig. 3** A-ROD enhances DKK1 transcription. **a** RT-qPCR levels of A-ROD, DKK1 and two unrelated long ncRNAs, in control and various siRNA treated cells, normalized to GAPDH and control condition. Whole-cell total RNA was reverse-transcribed with random primers. Error bars represent normalized standard deviations from three independent experiments ($n = 3$ biological replicates, $*p < 0.05$, $**p < 0.01$, $***p < 0.001$, two-tailed Student's $t$-test). **b** ChIP assaying total Pol II occupancy in control and A-ROD siRNA (#A-ROD.3'si2) treated MCF-7 cells using the N-20 antibody. Error bars represent standard deviations from three independent experiments ($n = 3$ biological replicates, $**p < 0.01$, two-tailed Student's $t$-test). **c**, **d** Chromosome conformation capture of **c** the DKK1 A-ROD interaction in control (EtOH) and 40 min E2 induced MCF-7 cells and of **d** the DKK1 A-ROD interaction in control (NC1) or A-ROD siRNA transfected MCF-7 cells, and in MDA-MB-231 cells. Relative interaction frequencies derived as described in Methods section

effect on the actual chromatin loop[2,18]. In order to assess the effects of A-ROD expression level and transcriptional changes on chromatin interactions to the DKK1 promoter we performed chromosome conformation capture (3C) in control and E2-treated MCF-7 cells. We also assayed another cell line, MDA-MB-231, where DKK1—but not A-ROD—is expressed (Supplementary Fig. 4a). The promoter of A-ROD in MDA-MB-231 cells is methylated (Supplementary Fig. 4f) and does not show characteristics of active transcription as opposed to MCF-7 cells where the promoter is hypomethylated and A-ROD is actively expressed (Supplementary Fig. 4a, e, f). While we can recapitulate the chromatin interaction between A-ROD and DKK1 in MCF-7 cells by 3C, E2-treatment does not affect it, despite transcriptionally repressing both A-ROD and DKK1 (Fig. 3c). Additionally, knockdown of A-ROD does not impede the interaction despite the effect on DKK1 expression (Fig. 3d), suggesting that the RNA does not have a role in establishing or maintaining the chromosomal looping. We also observe the same interaction between the DKK1 and A-ROD loci in the MDA-MB-231 cell line (Fig. 3d) despite no active transcription of A-ROD. These data argue that neither the RNA nor transcription affects this interaction, which is established prior to regulation mediated by A-ROD on DKK1.

**DKK1 is not regulated by chromatin-tethered A-ROD.** Our studies of chromatin-association of different groups of long ncRNAs suggest that dissociation from chromatin is important for the function of long ncRNAs engaged in strong long-range chromosomal interactions (Fig. 1). To address whether the regulatory effect of the long ncRNA happens while associated to chromatin, at or after dissociation from the chromatin-associated site of transcription, we employed transfection of both siRNAs and ASOs to deplete A-ROD, along with cellular fractionation and labeling of nascent RNA. Nucleoplasmic and chromatin-associated RNA was obtained from control and siRNA treated MCF-7 cells as described in[48,49] (Methods; Supplementary Fig. 6) and analyzed by RT-qPCR. While the chromatin-associated A-ROD RNA levels are poorly affected by siRNAs (Fig. 4a), the nucleoplasmic form is readily targeted (Fig. 4b) in agreement with recent studies showing assembly and function of the siRNA machinery in the nucleus[50]. Interestingly, upon A-ROD siRNA transfection we can recapitulate the decrease in steady-state DKK1 mRNA levels we see from whole cells (Fig. 3a) for the nucleoplasmic form of DKK1, but not for the chromatin-associated fraction (Fig. 4a, b, Supplementary Fig. 9a-b). To study the mechanistic details of this regulation, and while considering that downregulation of transcription may not be

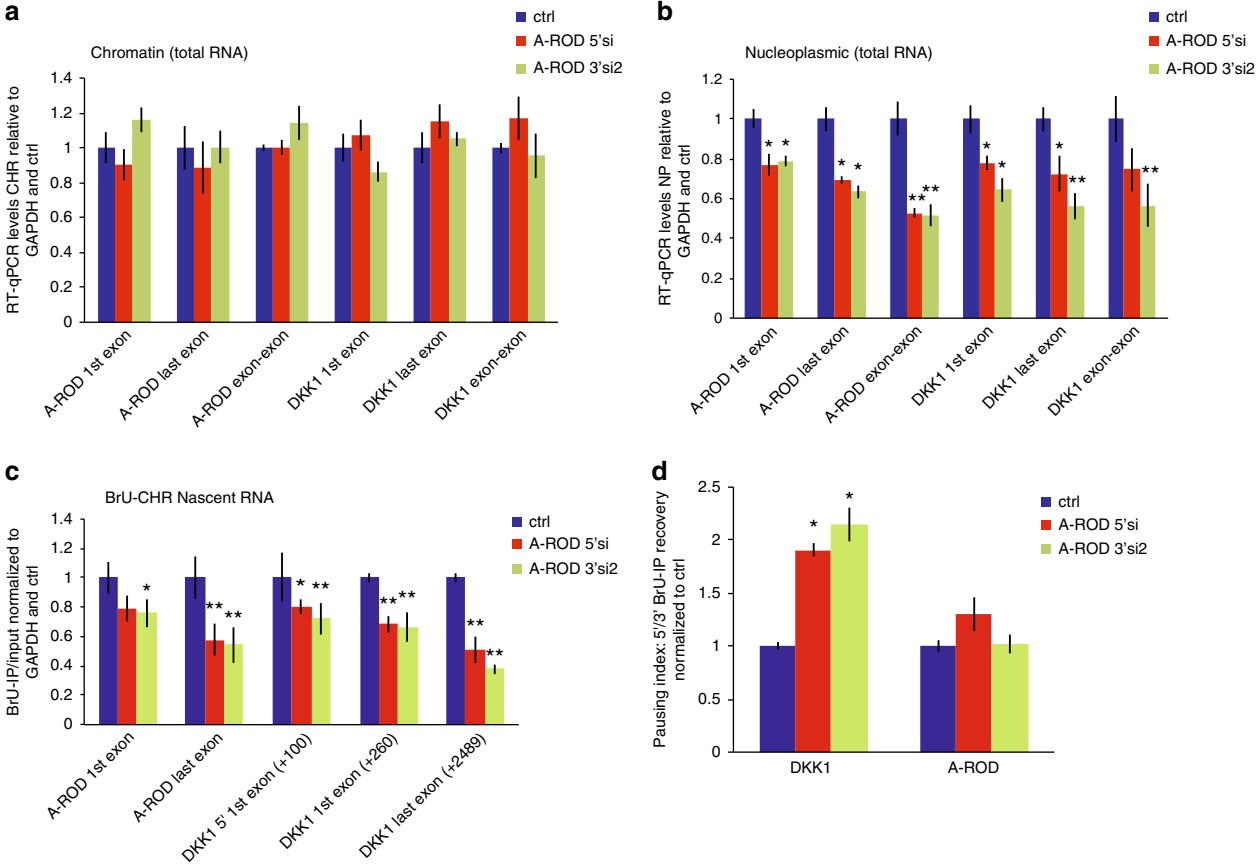

**Fig. 4** A-ROD is functional at its release from chromatin. **a, b** Expression levels of A-ROD and DKK1 assessed upon cell fractionation in the chromatin-associated (**a**) and nucleoplasmic fraction (**b**), in control (ctrl) and siRNA-treated MCF-7 cells. RT-qPCR measured RNA levels are normalized to GAPDH and control. Two different siRNA sequences targeting either the 5′ (A-ROD 5′si) or the 3′ (A-ROD 3′si2) of A-ROD were used. Error bars represent standard deviations from three independent experiments (n = 3 biological replicates, *p < 0.05 and **p < 0.01, two-tailed Student's t-test). **c** Expression levels of A-ROD and DKK1 in the BrU-labeled nascent RNA (BrU IP done from the chromatin-associated RNA fraction) measured by RT-qPCR in control and A-ROD siRNA treated cells, assayed by two different siRNA sequences targeting either the 5′ (A-ROD 5′ si) or 3′ (A-ROD 3′ si2) of A-ROD. Values were normalized to GAPDH exon and control. Error bars represent standard deviations from three independent experiments (n = 3 biological replicates, *p < 0.05 and **p < 0.01, two-tailed Student's t-test). **d** Transcriptional pausing index for A-ROD and DKK1 assessed by extracting the ratio 5′/3′ IP recovery (BrU-IP over Input for two distinct 5′ and 3′ RT-qPCR amplicons) in control and A-ROD siRNA treated cells

properly reflected in the steady-state chromatin-associated RNA, we purified nascent RNA from cells labeled by 5-bromouridine (BrU) incorporation into newly transcribed RNA for a short (15 min) pulse, followed by cellular fractionation[49]. We then analyzed the nascent (BrU-labeled) chromatin-associated RNA fraction to examine effects on nascent A-ROD as well as on transcription of DKK1. We observe a small effect of A-ROD siRNA on nascent A-ROD levels as determined by qPCR of the first exon, suggesting that the siRNA does not reduce the actual transcription at the A-ROD locus (Fig. 4c). There is, however, a greater reduction when quantifying the last exon, arguing that fully transcribed A-ROD can be better targeted by siRNAs while still loosely associated to the chromatin template, or at the very point of its release from chromatin. This result, recapitulated using siRNAs targeting either the 3′ or the 5′ of the A-ROD transcript, indicates that A-ROD is functional at its release from the chromatin-associated site of transcription, with A-ROD depletion causing a repressive transcriptional effect on DKK1 (Fig. 4c). Upon A-ROD knock-down there is a significant increase in the transcription pausing index[40,51,52] at the DKK1 locus (Fig. 4d), in agreement with A-ROD being involved in transcriptional enhancement of DKK1.

**DKK1 is regulated at A-ROD release from chromatin.** Taken together, the above results from the siRNA-mediated depletion of A-ROD suggest that the chromatin-associated form of A-ROD is not responsible for DKK1 transcriptional enhancement. To further substantiate that it is most likely the chromatin-released form of A-ROD exerting the regulatory effects, we used RNaseH-active antisense oligonucleotides (ASOs) targeting intronic sequences of A-ROD for chromatin-associated knock-down. We assayed four different intronic ASO sequences (Methods) that efficiently depleted chromatin-associated A-ROD at 16 h post-transfection (Fig. 5a, Supplementary Fig. 5a-b). We then pooled the two ASOs targeting the first intron of A-ROD (common intronic sequence for the two isoforms; Supplementary Fig. 5a-b) and the two ASOs targeting the second intron into single transfection experiments and assessed the steady-state chromatin-associated and nucleoplasmic RNA levels at different time points post-transfection (Fig. 5b, Supplementary Fig. 7a). In concordance with the single-oligo transfections (Fig. 5a), A-ROD levels appear significantly reduced in the chromatin-associated RNA fraction (Supplementary Fig. 7a). The steady-state nucleoplasmic levels remain unchanged in general and only at later time points show a small decrease (Fig. 5b). This is not surprising

given the ASOs targeting purely intronic sequences (Supplementary Fig. 5a–b) and the relatively high stability of A-ROD (Supplementary Fig. 4b). The small decrease observed in the steady-state nucleoplasmic levels of A-ROD caused by the intronic ASOs cannot account for the accompanying repressive transcriptional effect on DKK1, observed already at 16 h post

ASO transfection (Fig. 5c, d), similar to the effect exerted upon siRNA transfection (Fig. 4c, d), where the steady-state chromatin-associated levels of A-ROD remain unaltered (Fig. 4a). These results fit together in a model where what is actually depleted by the intronic ASOs is the active form of A-ROD at the point of its nascent transcript release from the chromatin-associated

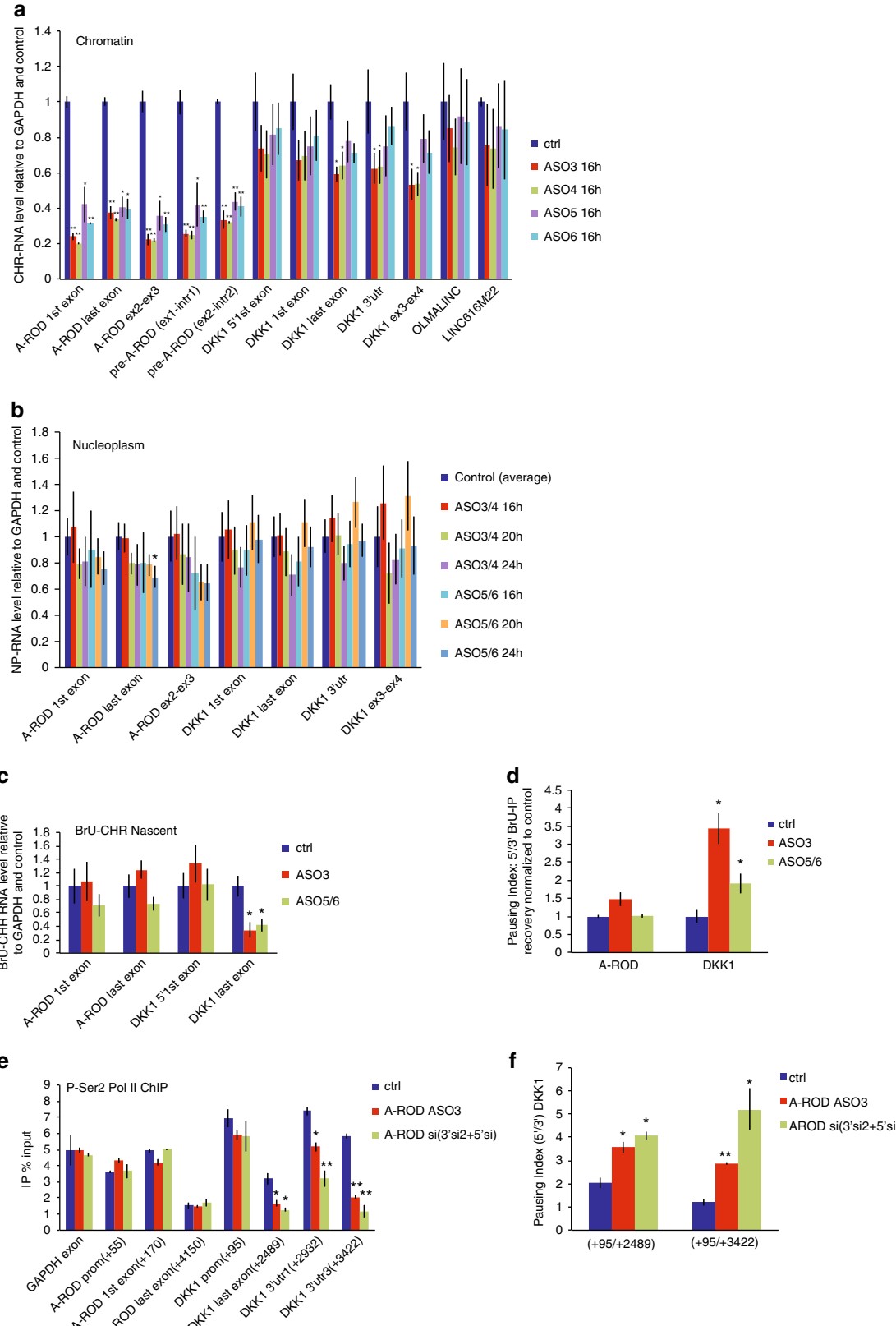

template. In a model where the intronic ASOs interfere with and impede the efficiency of co-transcriptional processing (splicing) of A-ROD, this would consequently result in reduced levels of the chromatin-dissociated and released (spliced) active form of A-ROD. To examine this possibility we performed BrU-IP from the chromatin-associated and nucleoplasmic fraction in control and 16 h ASO treated cells, and extracted the ratio of nascent nucleoplasmic to nascent chromatin-associated A-ROD (Supplementary Fig. 7b). The result shows that the intronic ASOs affect the release of the nascent A-ROD transcript, whereas there is no effect on nascent GAPDH transcript release. The reduction of nascent DKK1 release observed is likely due to the consequent transcriptional repressive effect on DKK1 (Fig. 5c,d) caused by the depletion of the active form of A-ROD. DKK1 transcription downregulation is likely exerted at the elongation step, as suggested by the increase in the pausing index (Fig. 5d). To further substantiate an effect on transcription elongation we performed ChIP of Pol II phosphorylated at Ser2 (P-Ser2 Pol II), which reflects the transcriptionally engaged RNA Pol II[51]. There is a significant decrease of P-Ser2 Pol II along the DKK1 body (Fig. 5e) and an accompanied increase in the pausing index (Fig. 5f), when knocking down A-ROD with either ASOs or a pool of siRNAs. Furthermore, A-ROD depletion does not alter the TFIIB ChIP signal at the DKK1 promoter, suggesting that it does not affect the formation of pre-initiation complexes (PIC) at the DKK1 promoter (Supplementary Fig. 7c). Taken together the above results support that A-ROD mediated transcriptional enhancement of DKK1 is exerted at the level of productive elongation rather than at the transcription initiation step.

**A-ROD retention at chromatin prevents DKK1 enhancement**. In addition to the siRNA and ASO approaches, we employed further methodologies to gain mechanistic insight into the A-ROD mediated DKK1 transcription regulation. Both 3′ end formation and splicing are necessary for efficient nascent transcript release from the chromatin-associated site of transcription[53–56]. By 3′RACE (Methods) we confirmed that A-ROD has one major cleavage and polyadenylation site (CPA) (Fig. 6a, Supplementary Fig. 5c-d). Just upstream of the CPA we find the hexamer AAUAAA (polyadenylation 'pA' signal or 'PAS'), and just downstream a T-rich stretch, suggesting that A-ROD follows the canonical 3′ end formation[57]. We used RNaseH-inactive 2′-O-methyl-phosporothioate RNA oligos (2′OMePS) to block PAS and/or CPA in 24 h transfection (Methods) aiming to interfere with A-ROD transcription termination and 3′ end formation, hence causing chromatin retention of A-ROD and impede its efficient nascent transcript release. Treatment with both blockers causes significant increase of A-ROD transcriptional read-

through, assessed by RT-qPCR on the nascent BrU-labeled chromatin-associated RNA (Fig. 6b) and by P-Ser2 Pol II ChIP (Fig. 6c), and a consequential impediment of DKK1 transcription (Fig. 6b, c). We also employed splicing-modifying morpholinos (MOs) (Methods) targeting the donor ('mo.sj3') and acceptor ('mo.sj4') splice sites of the second intron of A-ROD (Fig. 6a, d, Supplementary Figure 5a-c). To visualize the effect we analyzed PCR products of both the pre-mature (non-spliced) and mature (spliced) forms of A-ROD (Fig. 6d) and further assessed the splicing interference by RT-qPCR on chromatin-associated (Fig. 6e, f) and nucleoplasmic RNA (Fig. 6g). The mo.sj3 causes strong splicing inhibition, consequently resulting in a reduction of the steady-state nucleoplasmic RNA levels of A-ROD at 36 h post-transfection (Fig. 6g). Both the mo.sj3 and the less harsh mo.sj4 (Fig. 6d-f), which does not substantially reduce the steady-state nucleoplasmic A-ROD RNA levels (Fig. 6g), cause an accompanying transcription defect at the DKK1 locus (Fig. 6c). Together these results argue against a merely in trans regulation, and support that the nascent chromatin-dissociated spliced form of A-ROD is important for DKK1 transcriptional enhancement.

**siRNA-accessible A-ROD recruits EBP1 to the DKK1 promoter**. We further purified endogenous A-ROD RNA using antisense biotinylated oligos tiling the entire A-ROD transcript (Methods)[3,58,59]. Using probes targeting the A-ROD RNA we obtain a specific recovery of A-ROD compared to both control probes and recovery of DKK1 or GAPDH transcripts (Fig. 7a). To assess the binding of A-ROD RNA to the genomic region of DKK1 we analyzed the associated DNA retrieved under the same conditions from the affinity-purified RNA (Fig. 7b), showing that A-ROD associates to both the promoter and body of DKK1 but not to the GAPDH promoter as a control locus. In addition, we observe that A-ROD RNA binds to its own genomic locus, which likely reflects its active transcription. To examine whether A-ROD can recruit specific regulatory proteins to the DKK1 locus, we extracted the proteins associated to the affinity-purified A-ROD RNA and identified two candidates by mass spectrometry which show up specifically using the A-ROD antisense probes compared to the LacZ probes; the p48 isoform of EBP1 (ErbB3-binding protein 1) and HNRNPK (heterogeneous nuclear ribonucleoprotein K) (Fig. 7c). Western blot analysis confirms the specific enrichment of EBP1 by A-ROD (Fig. 7d), without recapitulating the specificity for HNRNPK, although this has been reported to bind specifically to some long ncRNAs[60,61]. The EBP1 isoform p48 has been reported to act as a context-dependent transcriptional activator[62] and is found to bind RNA in the nucleus[63]. To address whether EBP1 can bind to the DKK1 locus and to what extent this is dependent on A-ROD, we performed

---

**Fig. 5** Nascent A-ROD enhances DKK1 transcription elongation. **a** RT-qPCR assaying the steady-state chromatin-associated RNA levels of A-ROD, DKK1 and two unrelated transcripts, using several amplicons, at 16 h post-transfection of 50 nM ASOs targeting A-ROD intronic sequences (see also Supplementary Fig. 5a-b and Supplementary Fig. 7a). Error bars represent standard deviations from three independent experiments ($n = 3$ biological replicates, $*p < 0.05$ and $**p < 0.01$, two-tailed Student's $t$-test). **b** RT-qPCR assaying the steady-state nucleoplasmic RNA levels using several amplicons at three time-points post-transfection of pooled intronic ASO sequences transfected at 100 nM final concentration; either ASO 3 and 4 targeting the first intron of A-ROD or ASO 5 and 6 targeting the second intron; or the standard negative control ASO. Error bars represent standard deviations from three independent experiments ($n = 3$ biological replicates, $*p < 0.05$, two-tailed Student's $t$-test). **c** Normalized RT-qPCR levels in the BrU-labeled nascent chromatin-associated RNA (BrU IP done from the chromatin-associated RNA fraction) in control and A-ROD intronic-ASO treated cells. Error bars represent standard deviations from three independent experiments ($n = 3$ biological replicates, $*p < 0.05$, two-tailed Student's $t$-test). **d** Transcriptional effect assayed by extracting the 5′ to 3′ BrU-IP recovery ratio in control and A-ROD intronic-ASO treated cells. **e** ChIP for P-Ser2 Pol II in control MCF-7 cells, cells treated with siRNA against A-ROD (here double knock-down by cotransfecting pooled A-ROD.5′si and A-ROD.3′si2) and MCF-7 cells treated with the A-ROD intronic-ASO3 (here for 24 h). Error bars represent standard deviations from three independent experiments ($n = 3$ biological replicates, $*p < 0.05$ and $**p < 0.01$, two-tailed Student's $t$-test). **f** DKK1 transcriptional effect in control, A-ROD siRNA and intronic-ASO3 treated cells, assayed by extracting the ratio of 5′ to 3′ P-Ser2 Pol II ChIP signal depicted in (**e**)

ChIP for EBP1 with or without depletion of A-ROD using siRNA (Fig. 7e). We find that decreased A-ROD RNA levels lead to decreased EBP1 protein signal specifically at the DKK1 promoter, showing that the protein is actively recruited to the promoter of DKK1 by A-ROD. Here, GAPDH is used as a control locus where the promoter is bound by EBP1 but the interaction is not affected by A-ROD knock-down. While EBP1 shows a more general expression than A-ROD, the co-expression of DKK1, EBP1, and A-ROD is supportive of a regulatory relationship (Supplementary Fig. 4c). In addition, by knocking down EBP1 using siRNA in both MCF-7 (A-ROD positive) and MDA-MB-231 (A-ROD negative), we observe DKK1 reduction only in MCF-7 (Fig. 7f),

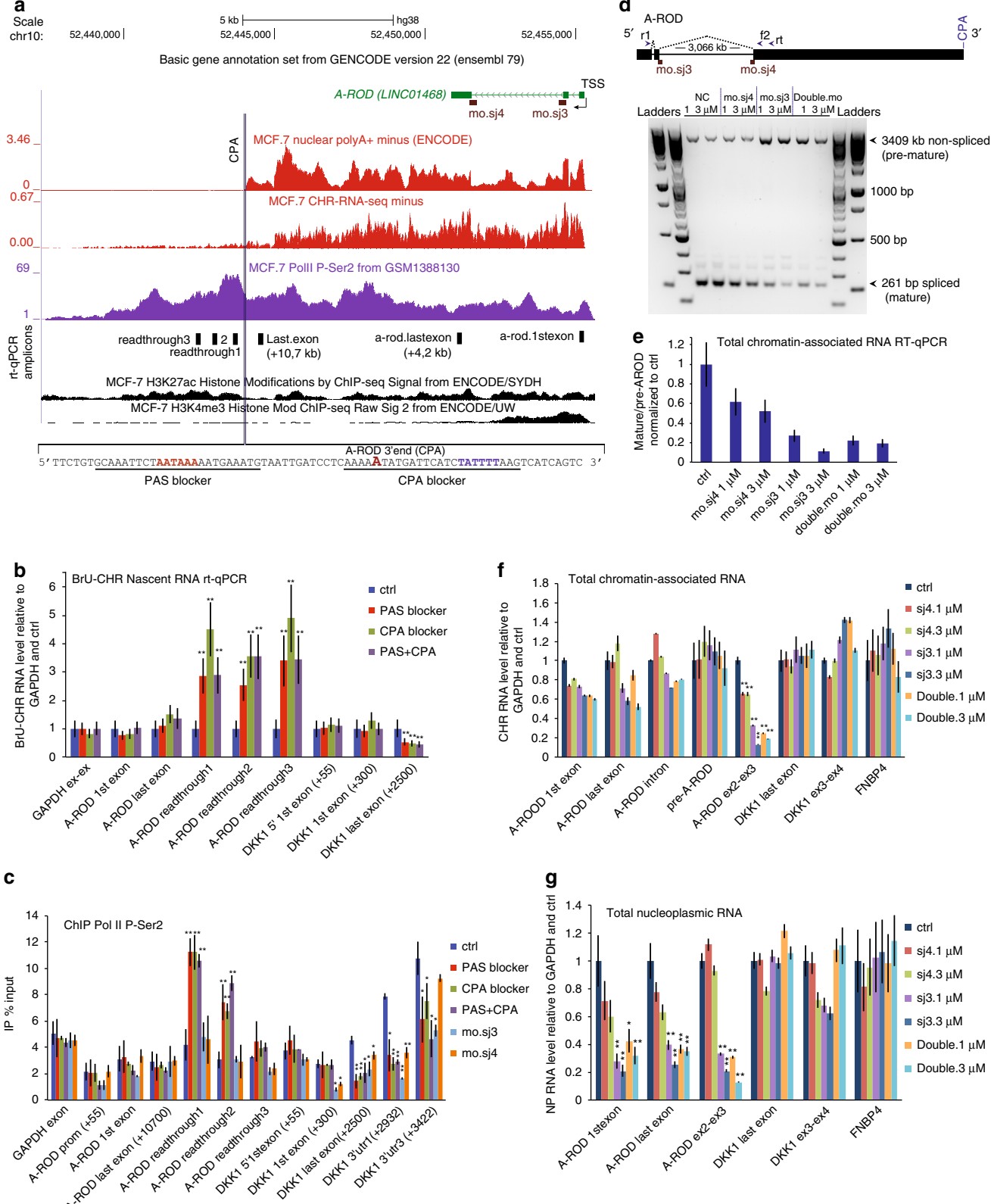

verifying A-ROD-mediated recruitment of EBP1 to the DKK1 promoter and that A-ROD enhances DKK1 expression in a context-specific manner.

## Discussion

Here, we group long ncRNAs by epigenetic marks and find that around a quarter are engaged in strong long-range chromosomal interactions in MCF-7 cells. While long ncRNAs are thought to be generally chromatin-enriched[6,36], our results show that those transcribed from loci engaged in strong long-range chromosomal interactions are enriched in the soluble nucleoplasmic fraction rather than remaining tightly chromatin-associated. The relatively greater release of long ncRNAs interacting with their target genes via pre-established physical proximity suggests that their dissociation from chromatin plays a role in their function. It could assist in recruiting regulatory protein factors to the transcription units of their target genes, with an optimal target specificity ensured within the pre-established chromosomal proximity (Fig. 8). On the other hand, in a recently studied case of a chromatin-associated long ncRNA, the regulation is exerted via a distinct mechanism, namely transcriptional interference of its antisense-overlapping gene[10]. In such cases of chromatin-associated long ncRNAs, while transcription activity is important, dissociation from chromatin does not seem to be a functional prerequisite, neither is the establishment of interactions via the formation of chromosomal loops.

We show that A-ROD, a spliced and polyadenylated long ncRNA, can positively regulate transcription of DKK1 at the level of transcription elongation and recruit the general transcription factor EBP1 to the DKK1 promoter. Based on our results, we propose that A-ROD associates with the DKK1 locus and recruits EBP1 to its promoter at its release from chromatin when it gets accessible to siRNA-mediated knock-down (Fig. 8).

How activating long ncRNAs expressed from enhancers can recruit proteins specifically to the promoters of their target genes is a question of high interest to the field. While only a few consensus motifs or little sequence complementarity have been found between target genes and long ncRNAs regulating their expression[64], the physical proximity is thought to be important in directing the long ncRNA to its target. How the chromatin interactions are established is not fully known, but based on our

and others' results this does not seem to be dependent on the transcription activity or the expression of the long ncRNA. We suggest, based on our results from siRNA- and ASO- mediated knock-down of nascent A-ROD and analysis in subcellular fractions, that the chromatin-associated A-ROD is not the active form required for transcriptional enhancement of DKK1. Rather, the RNA mediates its effect while loosely associated to chromatin or at its release from the site of transcription, where the A-ROD locus is already in proximity to DKK1. Our data support that A-ROD has to be fully transcribed in order to be accessible by siRNAs, which implies a functional effect of A-ROD on DKK1 transcription after A-ROD has been transcribed and at its release from chromatin. We therefore suggest a model where transcription of the long ncRNA is not the critical step but rather its dissociation from chromatin where we speculate that it becomes exposed to the nuclear environment and accessible to bind regulatory proteins (Fig. 8). Since this regulation occurs on the neighboring gene—brought into close physical proximity via the pre-established long-range chromosomal interaction—but requires the release of the long ncRNA from the chromatin-associated template, it would be in agreement with a quasi-*cis* mechanism of action where the targeting specificity is mediated and ensured by the chromatin interaction established prior to and independent of the expression of the long ncRNA, and the regulatory effect is mediated by the long ncRNA transcript at its release from chromatin.

Long ncRNAs expressed from enhancers are often polyadenylated and more stable than eRNAs that are mostly non-polyadenylated and short-lived and expected not to dissociate from chromatin[20–22]. While eRNAs have been shown to increase the affinity for DNA-binding proteins to their site of transcription[65], the active recruitment of proteins by spliced long ncRNA at release from chromatin could be one of the factors distinguishing these two groups of non-coding transcripts expressed from enhancers. The importance of dissociation from chromatin of long ncRNAs is a novel finding that should be explored further to establish its impact in targeted gene regulation and enhancer function.

## Methods

**Set of long ncRNAs and de novo transcript assembly.** Long ncRNAs used in this study are a combination of ENCODE V22 annotated long ncRNAs, previously

**Fig. 6** 3′end formation and splicing in A-ROD function. **a** UCSC genomic screenshot showing A-ROD expression tracks and positions of the assayed RT-qPCR amplicons. P-Ser2 Pol II ChIP-seq is from[74]. Sequences of the PAS and CPA target sites of 2′-O-MeRNA-PS oligo blockers are underlined. **b** Normalized RT-qPCR of BrU-labeled nascent chromatin-associated RNA in MCF-7 cells treated either with control or 2′-O-MeRNA-PS oligo blockers targeting the PAS and/or CPA site. Error bars represent standard deviations from three independent experiments ($n = 3$ biological replicates, **$p < 0.01$, two-tailed Student's $t$-test). **c** P-Ser2 Pol II ChIP recovery (% of Input) in control; MCF-7 cells treated with 2′-O-MeRNA-PS oligo blockers targeting the PAS and/or CPA site; and MCF-7 cells treated with morpholinos targeting either the donor splice site (mo.sj3) or the acceptor splice site (mo.sj4) of the second intron of A-ROD. Positions of the morpholinos are depicted in (**a**, **d**). Error bars represent standard deviations from three independent experiments ($n = 3$ biological replicates, *$p < 0.05$ and **$p < 0.01$, two-tailed Student's $t$-test). **d** Schematic representation of A-ROD exonic structure and PCR analysis of the premature (non-spliced) and mature (spliced) forms of A-ROD in control MCF-7 cells (treated with the 'standard control sequence' morpholino) or in MCF-7 cells treated either with the mo.sj3 or the mo.sj4 in the given concentrations. In the 'double' lanes both morpholinos were pooled and cotransfected in the given final concentrations. The PCR (30 cycles) was performed with primers r1 and f2 on cDNA reverse-transcribed from 200 ng total nuclear RNA primed with the 'rt' primer (see Methods section). Please note, that this PCR analysis is not quantitative and was conducted to visualize the effect of the morpholinos treatment. The accumulation of the premature product observed in lanes 5–8 (mo.sj3 and double.mo) is most probably due to the mo.sj3-exerted elimination of the mature form of A-ROD (hence less available spliced product to amplify) and both products being amplified by the same pair of primers, in the same reaction. To quantitatively assess the morpholino treatment effect, RT-qPCR analysis was further performed (**e-g**). **e** Normalized to control RT-qPCR measured abundance of the mature (spliced) form of A-ROD (amplicon 'exon2–exon3') divided to the RT-qPCR measured level of the premature (non-spliced) form (amplicon 'pre-A-ROD.2'), in control and in MCF-7 cells treated with morpholinos as in (**d**). The RT-qPCR values used to extract the ratio spliced/pre-mature A-ROD are from RT-qPCR on total chromatin-associated RNA (cDNA primed with random primers) depicted in (**f**). **f**, **g** RT-qPCR analysis on **f** total chromatin-associated RNA and **g** nucleoplasmic RNA (in both cases random primed cDNA) in control and MCF-7 cells treated with morpholinos as in (**d**). Error bars represent standard deviations from three independent experiments ($n = 3$ biological replicates, *$p < 0.05$ and **$p < 0.01$, two-tailed Student's $t$-test)

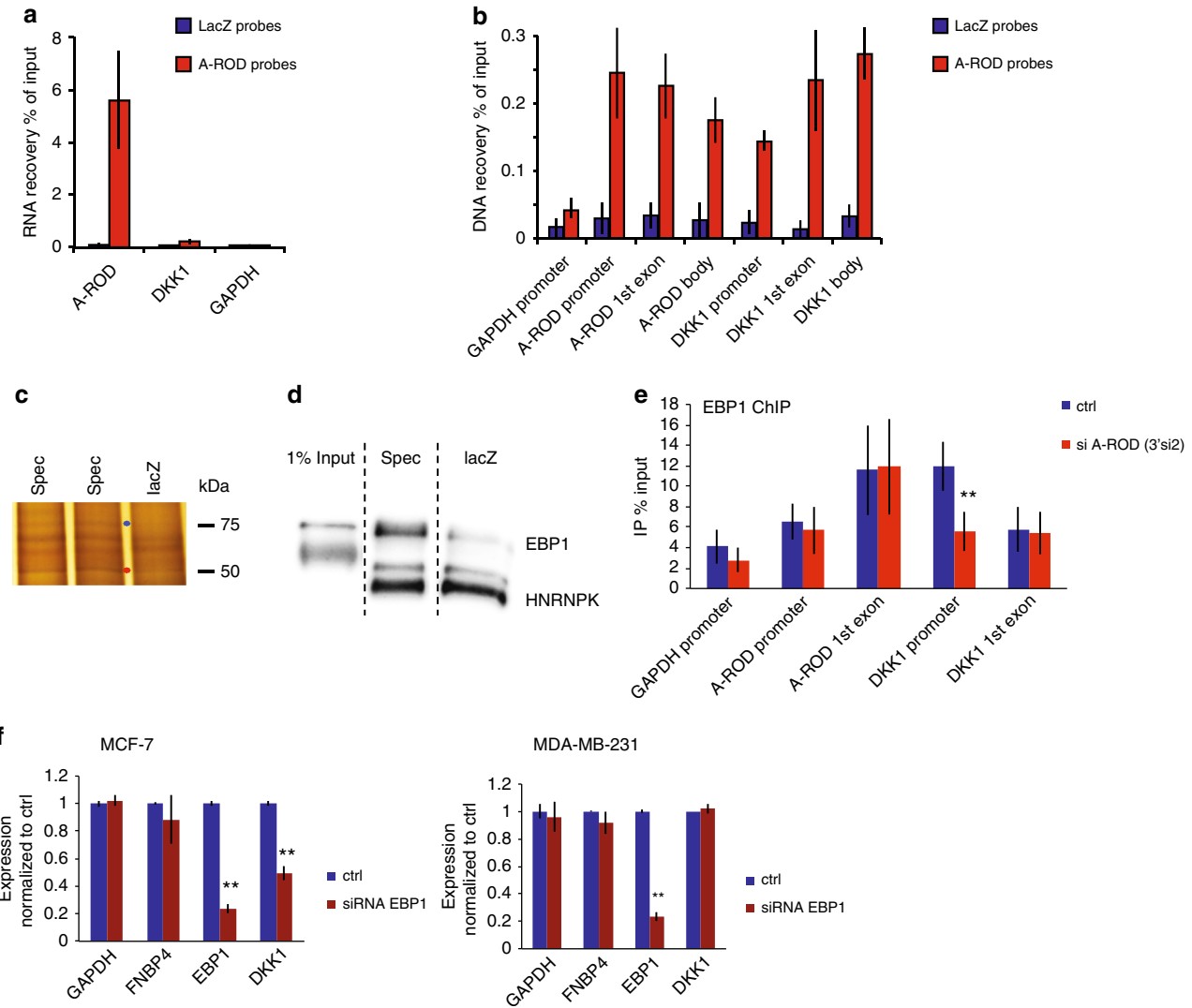

**Fig. 7** A-ROD recruits EBP1 to the DKK1 promoter. **a** RNA recovery of A-ROD, DKK1 or GAPDH transcripts after RNA purification, using biotinylated antisense probes targeting A-ROD (specific) or lacZ (control), measured by RT-qPCR. **b** DNA recovery of the depicted loci measured by qPCR of DNA eluted from RNA purification using biotinylated antisense probes targeting A-ROD or lacZ. Error bars represent standard deviations from three independent experiments ($n = 3$ biological replicates). **c** Silver staining of SDS-PAGE analyzed proteins, eluted from RNA purification using biotinylated antisense A-ROD specific or lacZ control probes. Marked bands were excised, destained and subjected to mass spectrometry. **d** Western blot for HNRNPK and EBP1 on SDS-PAGE analyzed proteins, eluted from RNA purification using biotinylated antisense A-ROD specific or lacZ control probes. Anti-HNRNPK was first applied on the blot, followed by anti-EBP1. **e** ChIP for EBP1 in control and A-ROD siRNA treated MCF-7 cells. Error bars represent standard deviations from three independent experiments ($n = 3$ biological replicates, **$p < 0.01$, two-tailed Student's t-test). **f** Normalized RT-qPCR RNA levels in control and EBP1 siRNA treated MCF-7 and MDA-MB-231 cells. Error bars represent standard deviations from three independent experiments ($n = 3$ biological replicates, **$p < 0.01$, two-tailed Student's t-test)

annotated long ncRNAs[32] and de novo transcript assembly using MCF-7 chromatin-associated RNA-seq data (available under GEO accession number GSE69507). In detail, chromatin-associated RNA-seq were mapped to GRCh38 (hg38) using STAR[66] (version 2.4.2.a) with default settings. De novo transcript assemblies produced by Cufflinks[67] (version v2.2.1) and StringTie[68] (version v1.1.1) were merged in the BED12 format using bedtools[69] -groupby, collapsed by name. All ENCODE V22 annotated transcripts were subtracted (using bedtools[69] -subtract) and the remaining non-overlapping transcripts were subjected to CPAT[33] to exclude transcripts with coding potential. Only transcripts with no coding potential and at least one splicing junction were kept which yielded 428 de novo predicted long ncRNA transcripts not included in any ENCODE annotation. Finally transcripts overlapping on the same strand were merged selecting for the very first TSS and the very last TTS giving rise to 341 distinct de novo predicted long ncRNAs. ENCODE V22 annotated long ncRNAs were filtered for no overlap with protein coding genes and monoexonic transcripts were discarded producing 7003 long ncRNAs (strand-specific merged from 8128 transcripts). Finally another set of previously annotated long ncRNAs[32] were filtered for no overlap with ENCODE V22 and no overlap with the 341 de novo predicted long ncRNAs, as

well as no coding potential (CPAT[33]) and at least one splicing junction, resulting in 3262 more long ncRNAs. From the 10,606 in total long ncRNAs, 4467 with GRO-seq[35] RPKM > 0 and chromatin-associated RNA-seq RPKM > 0 were subjected to the k-means clustering.

**ChIA-PET**. ChIA-PET reads in MCF-7 from 4 replicates[29,30] were pooled and filtered for a cutoff score of 200. Pairs of interacting long ncRNA with annotated ENCODE V22 genes were identified by intersecting intervals of ±2 kb around the TSS with the ChIA-PET nodes, using bedtools[69].

**Methylation status analysis**. We used the CpG Methylation data of Methyl 450 K Bead Arrays from ENCODE/HAIB. We used the Jensen-Shannon divergence of methylation levels across cell-types studied as described in[70] to extract the hyper-methylation and hypo-methylation specificity value per CpG using the 450 K Array defined b-values from a panel of 63 cell lines or subsets. The maximum CpG hypomethylation specificity value was assigned per promoter.

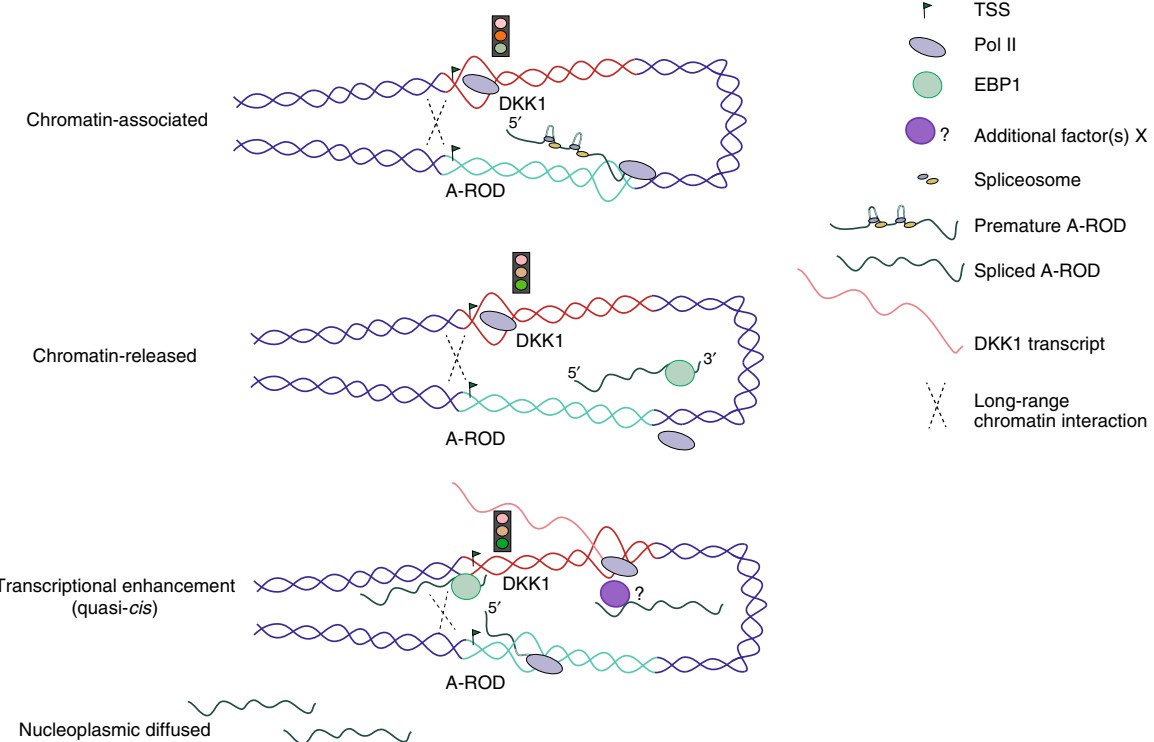

**Fig. 8** A-ROD enhances DKK1 transcription with a quasi-*cis* mechanism. Model for A-ROD-mediated recruitment of EBP1 and regulation of DKK1 transcription at release from the chromatin-associated template, and within the physical proximity of the pre-established chromosomal interaction

**RNA-seq and expression analysis**. Chromatin-associated RNA-seq libraries in MCF-7 cells were prepared as in[48,49] and mapping of strand-specific reads was done with the STAR algorithm[66]. Nucleoplasmic-enriched RNA-seq libraries were prepared from the nucleoplasmic fraction upon cell fractionation as in[48,49] and mapping was done using STAR[66]. Differential expression analysis of chromatin-associated vs. nucleoplasmic RNA as well of available GRO-seq data[35] (GEO accession number GSE43835) was performed using DESeq2[37]. Expression specificity was extracted using the same script for extracting CpG hypo-methylation specificity (see the Methods section Methylation status analysis), loading with strand-specific read counts from RNA-seq data[30].

**K-means clustering**. Peaks from available MCF-7 histone mark ChIP-seq data (ENCODE H3K4me3 and H3K27ac) and H3K4me1 from[34] (GEO GSM588569) were called at $p < 0.01$ using MACS2 (version 2.0.10.20120913)[71]. Signal values from long ncRNA overlapping peaks were summed across transcript unit length. All data values, including log transformed GRO-seq and CHR-RNA-seq RPKM, were rescaled in the range between 0 and 1 using min–max normalization. K-means clustering was performed with R function *kmeans*, with k = 15 (at this number the total within-cluster sum of squares reached a minimum plateau) and nstart = 100.

**Linear regression and PCA**. Linear regression was performed with R function lm on $Z$-score standardized values of the included parameters for long ncRNAs with more than 10 GRO-seq reads per kb. PCA was performed with R function prcomp and visualized with ggbiplot (https://github.com/vqv/ggbiplot). Points on the biplot represent projections of the original values in a new space, defined by a new set of variables, the principal components PC1 and PC2, which better explain the observed patterns in the data. Points that are clustered together correspond to data (in this case long ncRNAs) with similar values of their variables. Vectors indicate the direction and strength of each variable to explain the overall data distribution. The longer the vector (i.e. the bigger the absolute value of a variable), the higher the contribution of that variable to separate the data. The direction of each vector corresponds to the direction that better correlates with one or the other component, i.e., in our case all five variables positively correlate with the first principal component, which also explains most of the data variability. Vectors that point in the same direction correspond to variables that have similar response profiles and are positively correlated, i.e., H3K4me3 and ChIA-PET score are highly correlated in our case, and are also positively correlated to the score of nucleoplasmic-enriched long ncRNAs (most of the nucleoplasmic-enriched long ncRNAs have a PC1 score > 0 (right of the midpoint; Fig. 1f).

**Breast cancer RNA-seq data**. RNA-seq data from ER positive and ER negative breast cancer samples were downloaded from the Cancer Genome Atlas (TCGA; TCGA_BRCA_exp_HiSeqV2_exon_2014-08-28). Total exon RPKM (reads per kb per million) were summed per transcript.

**GTEx Project data**. Expression data from the Genotype-Tissue Expression (GTEx) Project were obtained from the GTEx portal, latest release version V6p.

**Bisulfite sequencing**. Bisulfite sequencing in MCF-7 cells was done using the SureSelect Methyl-Seq Target Enrichment System from Agilent. Data analysis (including read mapping, deduplication and methylation calling) was done using Bismark[72]. Data available under GEO (NCBI) accession number GSE69505.

**Purification of nascent RNA**. MCF-7 cells in culture, seeded 1 day before, were incubated with 2.5 mM final concentration of 5-bromouridine for 15 min, then harvested on ice and subjected to cell fractionation[49]. Thirty microliter of Protein G Dynabeads were incubated with 5 μg of purified mouse anti-BrdU antibody (BD Biosciences) for 1 h at room temperature. ~4 μg of BrU-labeled chromatin-associated or nucleoplasmic RNA were incubated with precoupled antibody (1 h at room temperature in the presence of RNase inhibitors) followed by five washes in ice cold PBS. Nascent RNA was eluted twice from the beads with competitive elution (25 mM 5-bromouridine in PBS) for 1 h at 4 °C. Eluates were pooled and precipitated with EtOH or using concentrator columns (Zymo Research). All of the eluted nascent RNA (and ~400 ng of chromatin-associated or nucleoplasmic input RNA) was subjected to reverse transcription with random hexamer primers. For increased resolution (Fig. 6b) the BrU-labeled chromatin-associated nascent RNA was fragmented (0.1 M NaOH, 20 min on ice) just before the BrU-IP followed by purification using concentrator columns (Zymo Research).

**Purification of target RNA and associated DNA and proteins**. Purification of target RNA and associated DNA and proteins was done based on the previously published methods ChIRP[59] and CHART[58] with modifications. Twenty two nucleotide long antisense oligos tiling the entire length of A-ROD transcript were designed using the ChIRP Probe Designer version 4.2 (Biosearch Technologies), biotinylated with TdT enzyme and Biotin-11-dUTP using standard protocol and purified from 10% PAGE. MCF-7 cells were crosslinked with 1% formaldehyde for 10 min, quenched with 0.125 M glycine and washed once with PBS. Fixed cells were lysed on ice in cell lysis buffer (10 mM Tris pH 7.5; 150 mM NaCl; 0.15% IGEPAL) and cell lysate was layered on top of 2.5 volumes 24% sucrose solution (10 mM Tris pH 7.5; 150 mM NaCl) according to the cell fractionation protocol[49]. Nuclei were pelleted at 7000 g for 10 min, shortly washed once with ice cold PBS and

crosslinked with 1 % formaldehyde for 15 min, followed by 0.125 M glycine quenching and PBS wash. Nuclei were lysed for 1 h on ice in nuclear lysis buffer (10 mM Hepes pH 7.6, 7.5 mM MgCl2, 0.2 mM EDTA, 0.3 M NaCl, 1 M Urea, 1% Igepal) followed by Bioruptor sonication 2 × 15 cycles 30 s on/30 s off. Debris was discarded by centrifugation and supernatant chromatin was resonicated to ~ 0.5–3 kb fragment sizes. 50 µg of chromatin was incubated with 500 pmoles of probes in Nuclear Lysis buffer:PBS 1:1 (final urea concentration 0.5 M) overnight at room temperature in the presence of protease and RNase inhibitors. Then, 2 % was removed to use as input and 100 µl Dynabeads Streptavidin C1/T1 (prewashed in 1 × Binding & Wash buffer; 1 M NaCl, 10 mM Tris 7.5) was added (30 min incubation). Beads were washed five times 5 min each in 1xBW buffer supplemented with 0.1% Tween-20 and 0.1% SDS. For nucleic acid elution beads were resuspended in 95% formamide and boiled at 95 °C for 5 min to disrupt streptavidin-biotin bond. Crosslinks were reversed 2 h at 65 °C (0.1% SDS; 0.2 M NaCl) followed by acidic phenol/CHCl3 for RNA extraction and neutral phenol/CHCl3 for DNA extraction, and EtOH precipitation. For protein analysis reverse crosslinking was done on the beads in 1 × LDS buffer (Invitrogen) for 2 h at 68 °C, eluted proteins were resolved on 10% SDS-PAGE and analyzed either by silver staining (Thermo Scientific), band excision, destaining and mass-spectrometry or by Western blot.

**Chromatin immunoprecipitation (ChIP).** Cells were cross-linked (1% formaldehyde 10 min at room temperature). Reaction was quenched (0.25 M glycine) 5 min at room temperature. Cells were lysed 10 min on ice in whole cell lysis buffer (10 mM Tris pH 7.5, 150 mM NaCl, 0.15% IGEPAL, 1 × protease inhibitor), nuclei were collected by centrifugation, resuspended and incubated for 30 min on ice in low SDS nuclei lysis buffer (5 mM Tris-HCL pH 8.0, 10 mM EDTA, 0.1% SDS) supplemented with protease inhibitor, and lysed with a Bandelin tip sonicator (25% power, 3 × 5 cycles of 5 s, separated by pauses on ice). Chromatin was then fragmented to an average length of 200–300 bp in a Bioruptor sonicator (3 × 10 cycles 30 s on/30 s off, high intensity). Samples were centrifuged to discard debris (20 min 13.200 rpm 4 °C), and 40 µg of chromatin were incubated with either 8 µg Pol II ab N-20 (Santa Cruz Biotechnology sc-899 X, discontinued), 4 µg a-EBP1 (Bethyl A303-084A or Abcam ab33613), 5 µg a-TFIIB (sc-225 X, C-18), 2 µg a-phospho-Ser2 Pol II (NEB E1Z3G) or the respective amount of affinity purified rabbit IgG control, rotating overnight at 4 °C in 1 ml ChIP dilution buffer (1.1% Triton X-100, 1.2 mM EDTA, 16.7 mM Tris-HCl pH 8.0, 167 mM NaCl). Thirty microliter Protein A Dynabeads (prewashed three times in ChIP dilution buffer) were added and incubated for 30 min at room temperature with rotation. Beads were washed three times in low salt buffer (0.1% SDS, 1% Triton X-100, 2 mM EDTA, 20 mM Tris pH 8.0, 167 mM NaCl) and two times in high salt buffer (0.1% SDS, 1% Triton X-100, 2 mM EDTA, 20 mM Tris pH 8.0, 250 mM NaCl). DNA was eluted twice in 2 × 150 µl elution buffer (0.5% SDS, 0.1 M NaHCO3) by shaking 2 × 15 min at 65 °C. ChIP eluates were supplemented with 5 mM EDTA and 0.2 M NaCl and cross-links were reversed for 5 h at 65 °C (or overnight), followed by Proteinase K (1 h at 45 °C) treatment. DNA was purified on QIAGEN columns and eluted in 300 µl TE. All ChIP experiments were done with at least three biological replicates. Values from the rabbit IgG control have been subtracted from the IP % Input recoveries.

**3′Rapid amplification of cDNA ends (3′RACE).** Total nuclear RNA was isolated from MCF7 cells as in ref.[49]. Dynabeads Oligo(dT)25 (NEB) were used to enrich for poly-A tailed RNA following the enclosed protocol. Fifty nanogram of poly-A+ enriched RNA was converted to first strand cDNA using SuperScript III (Thermo Fisher Scientific) and the primer AP_RT.3′RACE (please see Oligonucleotide Sequences in Methods). Two microliter of the cDNA product was used directly in PCR reactions with Taq DNA polymerase (95 °C 60 s and 32 cycles of 95 °C 30 s, 63 °C 20 s, 72 °C 20 s) followed by a final extension 72 °C for 10 min), and primer pairs either (AUAP and A-ROD.last_exon.R; 560 bp 3′ RACE product) or (AUAP and A-ROD.last_exon.pA_R; 180 bp 3′ RACE product). The final PCR products were analyzed on a 1% agarose gel. The correct size band was gel-extracted and sent for sequencing (Eurofins) for confirmation.

**2′OMePS transfection.** To interfere with A-ROD transcription termination and 3′ end formation, MCF-7 cells were treated with RNaseH-inactive 2′O-methyl phosphorothioate RNA oligos (2′OMePS) (IDT) at 500 nM final concentration, in 24 h transfection with Hiperfect (Qiagen).

  PAS blocker
  mC*mA*mU*mU*mU*mC*mA*mU*mU*mU*mU*mA*mU*mU*mA*mG*mA*mA*mU*mU*mU*mG*mC

  CPA blocker
  mC*mU*mU*mA*mA*mA*mA*mU*mA*mG*mA*mU*mG*mA*mU*mA*mU*mC*mA*mU*mA*mU*mU*mU*mU*mU

  ctrl
  mC*mC*mU*mC*mU*mU*mA*mC*mC*mU*mC*mA*mG*mU*mU*mA*mC*mA*mA*mU*mU*mU*mA*mU*mA

  * Phosphorothioate bond, m: 2′-O-Methyl RNA base

**Morpholinos.** To interfere with A-ROD splicing, MCF-7 cells were treated with splicing-inhibiting morpholinos (designed and purchased from GeneTools) at 1–3

µM final concentartion in 36 h transfection with Hiperfect (Qiagen). To increase uptake efficiency, the morpholinos were before transfection annealed to their DNA 'leash' (partially complementary DNA stretch) as described in[73]. Twenty five micromoles of the morpholino were mixed with 25 µmoles of the DNA leash in 250 µl 2.5 × PBS. The mixture was heated at 95 °C for 2 min, heatblock was turned off and let slow cooling down to room temperature to complete annealing. Final concentration of the double-stranded form is 100 µM. Standard control oligo sequence: CCTCTTACCTCAGTTACAATTTATA/ DNA leash tgatcGTAACT-GAGGTAAGAGGgtgat; mo.sj3 CACTCTCCTGGTTACCTCCATCTTC/ DNA leash tgatcAGGTAACCAGGAGAGTGgtgat; mo.sj4: MO sequence AGCCCTGGCTGATTCCAAAGATACA/ DNA leash gatagTGGAATCAGCCAGGGCTgtgat.

**Cell culture and siRNA/ASO knock-down.** MCF-7 cells were grown in DMEM (5% FCS) or in hormone-free medium (high glucose DMEM without phenol red) supplemented with 5% charcoal stripped FCS for 72 h before E2 induction. For siRNA mediated knockdown cells were transfected twice (36 h interval) with a 35 nM final concentration of siRNA. Cells were harvested at maximum 36 h after second transfection. Biological replicates with at least two different siRNA sequences were performed. For ASO knock-down of A-ROD, cells were transfected with a 50–100 nM final concentration of the intronic ASO and harvested after 16 h. SiRNA and ASO sequences are listed in Methods section.

**Cell fractionation.** Cell fractionation was done following a standardized protocol[49]. Briefly cells (from a confluent P10; ~9 × 10^6) were scraped in PBS and lysed in 400 µl buffer (10 mM Tris pH 7.5, 150 mM NaCl, 0.15% NP-40), then layered on top of 2.5 volumes 24% sucrose buffer and precipitated 5 min at 3500 g. The supernatant is the cytoplasmic fraction, whereas the pelleted nuclei were washed twice in 1 ml ice cold PBS and resuspended in 250 µl 50% glycerol buffer. An equal volume of nuclear lysis buffer (10 mM Hepes pH 7.6, 7.5 mM MgCl2, 0.2 mM EDTA, 0.3 M NaCl, 1 M Urea, 1 NP-40) was added and incubated on ice for 5 min followed by short vortexing and precipitation at full rpm for 2 min at 4 °C. The supernatant (nucleoplasmic RNA) was uptaken and the pelleted chromatin fraction was washed twice with ice-cold PBS. RNA was extracted from all fractions twice with acidic Phenol (pH 4.5) and once with CHCl3 and precipitated with three volumes of EtOH and 1/10 V NaAc pH 5.2.

**BrU-labeling pulse-chase.** To assess transcript turn-over rates, MCF-7 cells in culture were incubated with 2.5 mM final concentration of 5-bromouridine for 30 min, washed twice with PBS and further incubated in fresh medium containing 20 mM uridine for 0 min, 30 min, 1 h, and 4 h. Cells were harvested on ice and lysed for 5 min in 400 µl whole-cell lysis buffer (10 mM Tris pH 7.5, 150 mM NaCl, 0.15% NP-40). Whole cell total RNA was isolated by adding an equal volume of acidic phenol (pH 4.5), extracted twice with acidic phenol and once with CHCl3, followed by EtOH precipitation in the presence of 1/10 V NaAc pH 5.2. 0.5 µg of RNA from triplicate pulse-chase experiments were used in reverse transcription reaction with random hexameric primers, followed by qPCR.

**Chromosome conformation capture (3C).** 3C was performed as described in[4] with minor modifications. 10^6 cells were harvested and fixed in 20 ml PBS single-cell suspension (1% formaldehyde final conc.). Reaction was quenched (0.25 M glycine); cells were spun down and washed twice with cold PBS, resuspended in 10 ml cold cell lysis buffer (10 mM Tris pH 8.0, 10 mM NaCl, 0.2% Igepal, 1× protease inhibitor) for 30 min on ice, and lysed with a Wheaton Dounce tissue grinder (~10 strokes with pestle A, 10 strokes pestle B). Nuclei were collected (5 min at 2000 rpm) and washed with appropriate 1.2 × NEB buffer 3. Nuclei were resuspended in 500 µl 1.2 × NEB buffer 3 and treated with 0.3% SDS at 950 rpm for 1 h at 37 °C, followed by 1.8% Triton X-100 for 1 h at 37 °C. Eight hundred units of BglII were added and reactions were incubated overnight at 37 °C. Enzyme was inactivated with 1.6% SDS final concentration (25 min at 65 °C). Reactions were equilibrated in (1 × ligase buffer, 1% Triton X-100, 0.1 mg/ml BSA) in a final volume of 7.5 ml for 1 h at 37 °C. Reactions were chilled on ice and incubated with 1 mM ATP and 4000 units T4 DNA ligase (NEB) for 4 h at 16 °C. Reaction was stopped with 10 mM EDTA and cross-links were reversed overnight at 65 °C in the presence of Proteinase K (0.125 mg/ml). DNA was extracted with phenol-CHCl3 pH 8.0 and CHCl3 and precipitated (0.3 M NaOAc pH 5.2, 2.5 volumes ETOH). DNA was resuspended in 400 µl TE, subjected to sequential phenol-CHCl3 pH 8.0 and CHCl3 extractions and re-precipitated. DNA was resuspended in 200 µl TE and samples were centrifuged (20 min 13,200 rpm 4 °C) to discard SDS remnants, followed by column purification (QIAamp DNA). Primers for qPCR were designed using Primer3 (within 50–150 bp from BglII) with optimal parameters of length 23 nt and annealing temperature 63 °C. Primer efficiency was assessed on serial dilutions of total ligation products of BglII digested RPCI11.C BAC clone (ID 436D6) and experimental absolute values (2^{(-Ct)}) were normalized to Ct values for 5 ng BAC template ('relative interaction frequencies').

**Locus-specific DNA methylation analysis.** 2 µg DNA was bisulfite converted (EpiTect Bisulfite, Qiagen). PCR of amplicons spanning the A-ROD promoter was done with Taq DNA polymerase (Invitrogen) using primer sequences that do not

span CpG and bearing the bisulfite conversions (C>T in the forward primer and G>A in the reverse). The A-ROD promoter fragment (−300_+100 bp around the TSS) was amplified with primers (AGCACCGTACTCAGAGTCTA-CAAGGACTCA, AATCCCTGCCAAATGTAACTTACACTCTA) from untreated DNA and with (AGtAtyGTAtTtAGAGTtTAtAAGGAtTtA, AATCCCTaCCAAA-TaTAACTTACACTCTA) from bisulfite-treated DNA.

**Cell lines used in this study**. MCF-7 (ATCC HTB-22) and MDA-MB-231 (ATCC HTB-26).

**Statistical analysis**. *P*-values were extracted using unpaired Student's two-tailed *t*-test for all features with normal distribution; otherwise Wilcoxon-Mann-Whitney test was applied when indicated. *P*-values < 0.05 were considered statistically significant.

In figures depicting RT-qPCRs, the error bars represent standard deviations from the indicated number of independent experiments (*n* biological replicates). In the case of normalization of an amplicon A over an amplicon B (or over control) the final standard deviation (SD) was accordingly corrected using the type

$$SD_{norm} = SQRT\big((SD(A)/Average(A))^2$$
$$+(SD(B)/Average(B))^2\big) * Average(A)/Average(B).$$

**Primers, siRNAs, and ASOs**. Primer sequences can be found in Supplementary Table 1. Sequences of oligos used for A-ROD purification, siRNA and ASO sequences are available in Supplementary Table 2. For EBP1 siRNA mediated knock-down, predesigned siRNA sequences were ordered from IDT: #hs.Ri. PA2G4.13.3 (targeting exon 9) and #hs.Ri.PA2G4.13.2 (targeting exon 13).

**Data availability**. Chromatin-associated RNA-seq, total nucleoplasmic RNA-seq and DNA methylation Bisulfite-seq from MCF-7 have been deposited to GEO of NCBI under the accession number GSE69507.

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

## Acknowledgements

We thank Brian Caffrey for critical reading of the manuscript. E.N. has been funded by a postdoctoral fellowship from the Alexander von Humboldt Foundation. Work in the authors' laboratories is funded by the Alexander von Humboldt Foundation and the German ministry for research and education through the Sofja Kovalevskaja Award; the German Research Council (SPP-1738); and The Novo Nordisk Foundation Hallas Møller Stipend to U.A.V.Ø.

## Author contributions

E.N. and U.A.V.Ø. designed the research, interpreted data and wrote the manuscript. E. N. performed experiments and bioinformatic analysis, apart from the DNA hypo-methylation specificity code written by J.M. A.L. and J.L. performed experiments and interpreted data. A.M. supervised bioinformatic analysis and wrote the manuscript. All authors read and approved the manuscript.

## Additional information

**Competing interests:** The authors declare no competing interests.

