## [Peer Review File · Nature Communications]

Reviewers' comments:

Reviewer #1 (Remarks to the Author):

This paper presents a new concept termed "chromatin-release" and aims to demonstrate the functional importance of this process for the activating function of a long ncRNA they term A-ROD. Long ncRNAs are identified using previous data and de novo transcript assembly from chromatin-associated RNA sequencing data. Using ChiA-PET data from MCF-7 cells to identify long ncRNAs that are within regions that have long-range chromatin interactions, the authors group long ncRNAs into clusters based upon expression, histone marks H3K4me3, H3K4me1, H3K27ac, and the ChiA-PET maximum interaction score. One cluster contained high interacting scores as well as high levels of all three histone marks, and long ncRNAs in this group were found to be more enriched in the nucleoplasmic fraction rather than bound to chromatin. The authors go on to analyze previous data generated in response to estradiol in MCF-7 cells and how interacting gene pairs are regulated. One of the top candidates with highest fold change in response to E2 treatment (down-regulated) was lncRNA A-ROD and its interacting gene DKK1. There is a significant correlation between A-ROD and DKK1 expression across breast cancer samples, suggestive of a gene activation function of A-ROD on DKK1. siRNA knockdown of A-ROD resulted in decreased DKK1 expression and a decrease in nucleoplasmic DKK1 and A-ROD (but not chromatin bound), suggesting the effect of A-ROD on DKK1 expression occurs after its release from chromatin. A similar effect was seen when nascent A-ROD was depleted by ASO, supporting a role in the production on chromatin and release into the nucleoplasm as contributing to DKK1 activation. A proteomics approach was used to identify proteins interacting specifically with A-ROD, and the transcriptional activator EBP1 was identified. Binding of EBP1 to the DKK1 promoter was decreased upon A-ROD knockdown, suggesting that A-ROD may be important in recruiting EBP1 to DKK1 for efficient expression.

The concept that this manuscript explores, the mechanism of production and dynamic release of a lncRNA as it "hands off" a transcriptional activator for long range gene regulation is of potentially significant impact to the field. The means by which the authors use multiple datasets (most of which were previously-generated by others and publicly available) to identify candidates is impressive, though further description of methods in some cases would improve the manuscript's clarity. The experimental approaches in the second half of the manuscript are a good first attempt to make the correlation between the physical association and release from chromatin and A-ROD activator function. To increase the impact of this manuscript and support of the conclusions, a number of experiments are required, including: a more-direct manipulation of chromatin retention of A-ROD, a further test of the interaction between the lncRNA and the DKK1 locus, and a more-direct test of the importance of the EBP1 protein in DKK1 regulation.

Main Points

1. The term "Chromatin-release" needs to be more clearly defined earlier in the text. The abstract is confusing and could be re-worded to improve clarity. Instead of chromatin-release, "release from chromatin" may be easier for readers to interpret. It is also confusing to say that lncRNAs "engaged in strong chromatin interactions are less enriched at chromatin" without a definition of what it means to be "engaged in strong chromatin interactions". Without some re-wording the phrase seems counter-intuitive.

2. The siRNA and ASO experiments, while suggestive of a role for "chromatin release" are not a very direct test of the model. A way to significantly increase the impact of this manuscript would be to prevent chromatin release by mutation of the cleavage/polyadenylation signal in A-ROD to keep it retained on chromatin. This seems feasible, given the fact that A-ROD appears to have well-defined exonic structure and is polyadenylated.

3. The ChIRP/CHART result would benefit from a condition where the fraction of A-ROD that is released from chromatin is depleted by siRNA. This would address a caveat to this experiment, which is that the A-ROD locus is already in close three dimensional proximity to the DKK1 locus and oligo capture of the nascent transcript of A-ROD may mediate a more indirect interaction with the DKK1 locus. Observing a decrease in A-ROD interaction at the DKK1 locus upon siRNA or ASO knockdown of A-ROD would confirm that recovery of the DKK1 locus is not due to three dimensional proximity with the A-ROD locus.

4. To more-directly demonstrate a relationship between EBP1, delivered by "released" A-ROD, and DKK1 expression, EBP1 should be knocked down in both MCF-7 and MDA-MB-231 cells to test that only in the presence of A-ROD (MCF-7s) is EBP1 important for DKK1 expression. Without this, the

fact that MDA-MB-231 cells express DKK1 in the absence of A-ROD call into question the context where A-ROD is important for DKK1 expression.

5. Another means to increase the impact of this paper would be to further validate the computational pipeline by demonstrating a similar mechanism for an E2-activated lncRNA-mRNA pair. This would be particularly impactful because it is not actually clear what regulates A-ROD transcription itself.

6. Statistics are inconsistently presented. Most are shown in figure legends at beginning (i.e. Fig 2E – but these statistics are incorrectly mentioned in text line 168 and should include P-values for both comparisons or $P < \text{the lower P-value}$). Figures 4,5,6,7,8 have stars but no P-values shown anywhere. It is important to note statistical test used and P-values for each figure.

7. Multiple editing issue exist in the manuscript. For some reason, every time a word should contain "ev", an "S" is in its place. Also, multiple notes from the authors to themselves are still in the manuscript. In most cases, these notes suggest changes that have not been made yet, but would improve the manuscript.

Minor Points

1. Figure 1D needs better brackets to show clusters and a designation of "ChIA-PET" in the MaxScore column.
2. The PC analysis in Figure 2C could do with further description to highlight its value. Explanation of some of the trendlines that the data fall along would help, for example.
3. The RT-qPCR analysis in Figure 4A is difficult to see in detail, and it is unclear which bars the stars of significance are supposed to mark. Some of these results could go in supplemental or be split into multiple figure panels.
4. It is unclear which GRO-seq data is used to filter expressed lncRNAs in MCF-7 cells.
5. The last two paragraphs of Results section should be moved to beginning of discussion.

Reviewer #2 (Remarks to the Author):

The major claim of this paper is that a lncRNA A-ROD needs to be released from chromatin in order to activate its target gene. This is indeed a novel claim and will contribute widely to our understanding of lncRNA function.

However, as written the manuscript is preliminary, in places rather ambiguous and I am not convinced that the data completely support the conclusions. This may be due to bad sentence construction as well as poor descriptions of methods and results which makes for difficult reading overall and making strong claims before the final bit of data is revealed. In order to communicate these data effectively to a wider community the manuscript requires substantial rewriting.

For me the exciting part of this work is that despite the close proximity of DKK1 and A-ROD brought about by chromatin conformation, the association is not influenced by expression of the lncRNA and that this confirms the work of others that have shown that chromatin conformation in many cases serves as a scaffold for regulatory interactions. (Intrachromosomal interactions have already in 2008 been described as scaffolds to recruit regulatory complexes Li et al 2008 MCB) I am also intrigued by the finding that the A-ROD transcript recruits EBP1 to the DKK1 promoter in trans to influence transcriptional elongation. These discoveries are what made me stick with reading this manuscript (many distracting typos and edits that have not been removed prior to submitting). Rather than the big focus on the "chromatin-release" the focus of the paper should centre on the EBP1 recruitment and the role an enhancer plays on transcript elongation as opposed to activation. The novelty here is how enhancing lncRNAs work (distinct from activating lncRNAs!)

Starting with the abstract - "chromatin-release" jumps out as a term which is not defined and could mean almost anything - the first thing I thought of was "releasing" chromatin looping associations between promoters and enhancers. It is only after the end of the 1st paragraph in the introduction that I began to understand that this work was examining whether the association of a lncRNA with chromatin was important for its role in regulating its target gene.

Introduction - enhancers, e-RNAs and activating lncRNAs are relevant, but this space could be better used to provide more background on the frequency of lncRNA associations/accumulation with chromatin at the site of transcription and some indication of what "chromatin-release of lncRNA" means: Does it infer that the lncRNA is associated with the chromatin post transcriptionally and then released, or does it mean the lncRNA is released into the nucleoplasm during transcriptional elongation? I prefer the terms chromatin associated lncRNA and nucleoplasmic lncRNA rather than released/disassociated lncRNAs

Results

Expression and chromatin-release: This section is overly long should be rewritten and reordered - I am not sure whether the E2 induction of transcription is required in this section it disrupts the flow.

The K-means cluster analysis should be described in the order of 1. De novo transcript assembly of chromatin-associated (i.e the author's own data for lncRNAs identified after fractionating chromatin from nucleoplasm in MCF7 cells) and how it is filtered to produce a short list of 4,467 lncRNAs for further analysis 2. Summary of k-means clustering analysis (Fig 1A and D together with published data (GRO-seq data from Hah et al?) and ChiA PET PolII and promoter/enhancer histone signatures, followed by figs 1E, 2A, and 2B and C (2D and E can go into suppl data. The data so far only provide evidence that "cluster 6" lncRNAs with the highest promoter/enhancer and PolII Chia-PET score enrichments have higher nucleoplasm enrichment than chromatin association. I am struggling to follow the argument of chromatin-dissociation here despite the linear regression analysis - a lack of chromatin association does not mean "chromatin release" . Perhaps more careful analysis of the GRO-seq data may indicate whether lncRNAs at the point of transcription associates with chromatin or is nucleoplasmic - and would help with understanding whether the lncRNA is transiently associated or not. I certainly can't see how the data show a "relatively higher release" (line 148).

Regardless of the semantics here, Chia PET interactions with nucleoplasmic lncRNAs as opposed to chromatin associated lncRNAs does not support the statement that chromatin-release is important for functions of these lncRNAs line 169- 171.

Line 173 -- this heading should read A-ROD and DKK are coordinately active in several cell lines. The conclusive sentence here line 195 is indicative of co-regulation not supportive of a tissue-specific activator.

Line 197 - this heading implies that A-ROD activates DKK1 transcription but the data below only shows a conclusive effect transcriptional elongation (not activation).

Line 211 "lncRNA target specificity" has nothing to do with the chromatin conformation data provided below. Further genomic information regarding enhancer and promoter elements as well as CTCF sites would be useful information for the reader to orientate the proximity interactions. In Fig 4C and D - why is the f2 site so variable between the MCF7 cells?

Line 238 As the authors have just concluded based on the siRNA experiments in Fig 4D, that the lncRNA is not involved in structuring the chromatin conformation, chromatin release (or otherwise) can't be important here.

A piece of very important data is required? What is the actual Chromatin:Nucleoplasmic ration for A-Rod? Is the lack of effect of siRNA on chromatin (Fig 5A) due to low amounts of chromatin associated RNA and therefore beyond the resolution of detection?

Why is the kd less efficient for A-ROD and DKK1 in Fig 5B compared to Fig 4A?

Rephrase lines 248 - 252 - there is no evidence (at this stage of the narrative) for whether the siRNAs are targeting elongation.

Fig 5C does not show whether the nascent RNA was isolated from different cell fractions! In some cases siRNA located within 100 bp of the TSS may prevent transcription initiation-elongation

(Stojic et al 2016 Nat. Comms). In the data presented here, this is not the case and that both the 3' and 5' siRNAs more effectively target A-ROD post transcriptionally or perhaps during elongation - NOTHING to do with the chromatin association.

Fig 6A - "data for DKK1"? Editing instructions still stuck in the ms? See also line 135 and 168! and 184!

At Fig 6, is where the data is interesting! The ASO's targeted to the intronic regions reduces chromatin associated A-ROD but not nucleoplasmic A-ROD and still has an effect on nascent DKK1 expression (however, not at the activation stage, since the 1st DKK1 exon is transcribed! and also supported by the pausing index experiment and later by the Ser2 Pol II ChIP). Fig S6A is important here - However, if ASO is preventing the release of A-ROD from the chromatin, why then is there no difference in A-ROD levels in the nucleoplasmic fraction? Is this because there is such a small amount of Chromatin associated A-ROD that it doesn't impact upon the levels of nucleoplasmic transcript?

What effect would ASO's targeting exons or siRNAs targeting introns have on these experiments?

I would have liked to have seen RNA Fish experiments with intronic and exonic probes to show nuclear and chromatin accumulation of A-ROD in addition to the CHIRP and CHART data in fig 8. Fig 8C is not particularly helpful, except to see that there were clearly more than 2 candidates.

If EBP1 is being recruited to the 1st exon of A-ROD genomic sequence (Fig 8E) then this does not match the model in Fig 8F. The conclusions would be strengthened and indeed validated by knock down experiments of EBP1 to see the effects on both transcripts as well as chromatin conformation.

If A-ROD regulates DKK1 in trans as proposed, would we not expect ectopically expressed A-ROD to enhance DKK1 expression?

The best part of the manuscript is the discussion which seems to have been written by a different author than the one who wrote the abstract, intro and results. However can the authors speculate how EBP1 binding to the promoter of DKK1 affects its transcriptional elongation.

Reviewer #3 (Remarks to the Author):

In this manuscript, Ntini et al. investigated the nuclear localization of long non-coding RNAs (lncRNAs) in MCF-7 breast adenocarcinoma cells and found that lncRNAs associated with strong chromatin interactions are less enriched in the chromatin fraction compared to other lncRNAs, which suggests a functional role in the release of lncRNAs from chromatin. The authors then focused on the A-ROD lncRNA and showed that the lncRNA does not mediate the chromatin interaction between the DKK1 gene and itself, but regulates transcription elongation of DKK1 after chromatin release instead.

On the whole, the manuscript feels more like a working draft rather than a complete piece of work. There are numerous comments strewn throughout, hinting at missing pieces of information. There is also a text substitution of "ev" into "S" throughout the entire manuscript. I would recommend the authors to use "lncRNA" as an abbreviation rather than "long ncRNA", especially since there is no mention of other non-coding RNA elements. The authors should also clearly note the datasets generated by themselves, and properly cite the public datasets used in the study.

1) The authors commented that there was a lag in the time response of lncRNAs compared to their target genes (Figures 1B and C), but the figures show the opposite trend. Furthermore, they compared Figures 1B and 1C while using linear and log Y-axis scales respectively. If both graphs are converted to linear Y-axis scales, there would arguably be a larger lag in the upregulated lncRNA-gene pairs instead of the smaller lag stated. Can the authors also provide the number of lncRNA-gene pairs considered in both graphs?

2) Can the authors show in detail how the chromatin-associated RNA Seq data was normalized to the nuclear poly-A+ RNA Seq data (Figure 2B)? I am uncertain about using RPKM ratio to compare the abundance of chromatin associated RNA and nuclear RNA. RPKM is an approximation of the RNA concentration within the sample (Wagner GP et al., *Theory Biosci.*, 2012), but the original amount of RNA from both fractions would have differed quite significantly. As a reference, Werner

et al. (Nat. Struct. Mol. Biol., 2017) compared the different fractions using spike in standards and absolute read counts.

3) Can the authors mention in detail how A-ROD does not fit the characteristics of an enhancer-associated eRNA? (line 185-186)

4) Can the authors show the expression of A-ROD and DKK1 (Figure 3B) in other breast cancer cell lines, such as T47D (ER+) and MDA-MB-231 (ER-), especially since MDA-MB-231 is used in the subsequent analyses?

5) What is the proportion of A-ROD and DKK1 RNA in chromatin, nuclear and cytoplasmic fractions? Can the authors also show the siRNA knockdown of A-ROD and DKK1 in the chromatin, nuclear and cytoplasmic fractions, normalized to a single control? It may be that the low siRNA efficiency in the chromatin fraction is partly due to the low abundance of target RNA in that fraction (Hu X et al., Nucleic Acids Res., 2004) (Hong SW et al., Nucleic Acid Ther., 2014) (Lennox KA and Behlke MA, Nucleic Acids Res., 2016)

6) I am not comfortable with the use of GAPDH as the reference gene for RT-qPCR in the different fractions of RNA. Since GAPDH is a protein coding gene, I would assume that its expression is not consistent between the different fractions. Can the RT-qPCR analysis be performed using one or more known chromatin-associated RNA as reference instead (with consistent expression in both chromatin and nuclear fractions)?

7) Why did the authors decide not to show the methylation status of the A-ROD promoter in MCF-7 and MDA-MB-231 cells? (line 222)

8) Which siRNAs were used in Figure 4D? Will it be possible to show the knockdown effects on MDA-MB-231 as well?

Reviewers' comments:

Reviewer #1 (Remarks to the Author):

This paper presents a new concept termed "chromatin-release" and aims to demonstrate the functional importance of this process for the activating function of a long ncRNA they term A-ROD. Long ncRNAs are identified using previous data and de novo transcript assembly from chromatin-associated RNA sequencing data. Using ChiA-PET data from MCF-7 cells to identify long ncRNAs that are within regions that have long-range chromatin interactions, the authors group long ncRNAs into clusters based upon expression, histone marks H3K4me3, H3K4me1, H3K27ac, and the ChiA-PET maximum interaction score. One cluster contained high interacting scores as well as high levels of all three histone marks, and long ncRNAs in this group were found to be more enriched in the nucleoplasmic fraction rather than bound to chromatin. The authors go on to analyze previous data generated in response to estradiol in MCF-7 cells and how interacting gene pairs are regulated. One of the top candidates with highest fold change in response to E2 treatment (down-regulated) was lncRNA A-ROD and its interacting gene DKK1. There is a significant correlation between A-ROD and DKK1 expression across breast cancer samples, suggestive of a gene activation function of A-ROD on DKK1. siRNA knockdown of A-ROD resulted in decreased DKK1 expression and a decrease in nucleoplasmic DKK1 and A-ROD (but not chromatin bound), suggesting the effect of A-ROD on DKK1 expression occurs after its release from chromatin. A similar effect was seen when nascent A-ROD was depleted by ASO, supporting a role in the production on chromatin and release into the nucleoplasm as contributing to DKK1 activation. A proteomics approach was used to identify proteins interacting specifically with A-ROD, and the transcriptional activator EBP1 was identified. Binding of EBP1 to the DKK1 promoter was decreased upon A-ROD knockdown, suggesting that A-ROD may be important in recruiting EBP1 to DKK1 for efficient expression.

The concept that this manuscript explores, the mechanism of production and dynamic release of a lncRNA as it "hands off" a transcriptional activator for long range gene regulation is of potentially significant impact to the field. The means by which the authors use multiple datasets (most of which were previously-generated by others and publicly available) to identify candidates is impressive, though further description of methods in some cases would improve the manuscript's clarity. The experimental approaches in the second half of the manuscript are a good first attempt to make the correlation between the physical association and release from chromatin and A-ROD activator function. To increase the impact of this manuscript and support of the conclusions, a number of experiments are required, including: a more-direct manipulation of chromatin retention of A-ROD, a further test of the interaction between the lncRNA and the DKK1 locus, and a more-direct test of the importance of the EBP1 protein in DKK1 regulation.

Main Points

1. The term “Chromatin-release” needs to be more clearly defined earlier in the text. The abstract is confusing and could be re-worded to improve clarity. Instead of chromatin-release, “release from chromatin” may be easier for readers to interpret. It is also confusing to say that lncRNAs “engaged in strong chromatin interactions are less enriched at chromatin” without a definition of what it means to be “engaged in strong chromatin interactions”. Without some re-wording the phrase seems counter-intuitive.

We thank the reviewer for this comment. We try to clarify this point, and replaced already in the abstract the phrase “long ncRNAs engaged in strong chromatin interactions” with “long ncRNAs *transcribed from loci* engaged in strong *chromosomal* interactions”. The “engagement” in strong chromosomal interactions refers to the chromosomal looping captured by Pol II ChIA-PET, as this has been further clarified in the text (lines 77-81). The long ncRNAs transcribed from such loci are *less enriched* in the chromatin-associated RNA fraction, meaning that their RNA levels or abundances are lower at chromatin, compared to long ncRNAs not transcribed from loci showing strong chromosomal interactions.

2. The siRNA and ASO experiments, while suggestive of a role for “chromatin release” are not a very direct test of the model. A way to significantly increase the impact of this manuscript would be to prevent chromatin release by mutation of the cleavage/polyadenylation signal in A-ROD to keep it retained on chromatin. This seems feasible, given the fact that A-ROD appears to have well-defined exonic structure and is polyadenylated.

We thank the reviewer for this excellent suggestion to gain more direct mechanistic insight. Indeed, A-ROD has a well-defined exonic structure, predicted by nuclear polyA+ RNA-seq data (ENCODE) and *de novo* transcript assembly from our steady-state chromatin-associated RNA-seq data (Supplementary Fig. 5). With PCR we confirm two splicing isoforms, one major and one minor expressed (Supplementary Fig. 5c). By 3'RACE we confirm one major cleavage and polyadenylation site (CPA) (Supplementary Fig. 5c-d), suggesting that A-ROD follows the canonical transcription termination pathway and 3'end formation (Ntini et al., NSMB 2013). Just upstream (-31 bp) of the CPA we find a canonical polyadenylation signal ('pA signal' or 'PAS'; the hexamer AATAAA) and a T-rich stretch just downstream of the CPA (Supplementary Fig. 5d). We followed the reviewer's suggestion by disrupting A-ROD 3'end formation using transient transfection of 2'-O-MeRNA-phosphorothioate RNaseH-inactive oligos (2'OMePS, IDT). These oligos hybridize to complementary sequences and cause steric interference and inhibition of regulatory interactions. We treated the cells with 2'OMePS oligos blocking the A-ROD pA signal and/or the CPA (Supplementary Fig. 5d) for 24 h (transfected at 500nM-1uM final concentration). Using both rt-qPCR on BrU-labeled fragmented chromatin-associated nascent RNA and Pol II P-Ser2 ChIP, we find that both the PAS and the CPA blockers impede A-ROD efficient transcription termination and cause read-through at the A-ROD locus (Figure 7). This 24 h treatment with the PAS/CPA blockers does not reduce transcription *per se* at the A-ROD locus, and causes an accompanying repressive transcription elongation effect at the DKK1 locus, again supporting that the chromatin-dissociated A-ROD is functionally important for DKK1 transcription regulation. To further substantiate that the chromatin-dissociated spliced product of A-ROD is the regulatory and functional form within the pre-established chromosomal proximity, we employed transient transfection of splicing-modifying Morpholinos (MOs) (GeneTools) to interfere with co-transcriptional A-ROD splicing and hence impede the release from chromatin of the nascent spliced A-ROD transcript. We used two morpholinos (and a 'standard' negative control) designed against either the A-ROD 2nd intron splice donor site ('mo.sj3') or the acceptor site ('mo.sj4') (Supplementary Fig. 5 a-c, Figure 8, Figure 7a,c). To visualize that MOs interfere with splicing (when annealed to their DNA 'leash' and transfected at 1 and 3uM final concentration for 36 h, please see Methods), we analyzed PCR amplified products (30 cycles) of both the pre-mature (non-spliced) and mature (spliced) forms of A-ROD (Figure 8a-b), and further assessed the splicing-interference effect by rt-qPCR (Figure 8c-e). The mo.sj3 (targeting the 2nd intron splicing donor site) turned out to be particularly efficient in causing splicing inhibition, consequently resulting in a reduction of the steady-state nucleoplasmic RNA levels of A-ROD at 36h post-transfection (Figure 8e). Both treatment with the mo.sj3 and the mo.sj4 (targeting the 2nd intron splice acceptor site; not as harsh as the mo.sj3 (Figure 8b-d), and not substantially reducing the steady-state nucleoplasmic A-ROD RNA levels (Figure 8e) cause an accompanying transcription elongation defect at the DKK1 locus (Figure 7c).

We should also note here that we have repeated the siRNA and ASO experiments with several different sequences, and assessed the expression from the chromatin-associated and

nucleoplasmic fraction at different time points post-transfection to support our results (Figures 5-6, Supplementary Fig. 5a-b, Supplementary Fig. 7a, Supplementary Fig. 9).

3. The ChIRP/CHART result would benefit from a condition where the fraction of A-ROD that is released from chromatin is depleted by siRNA. This would address a caveat to this experiment, which is that the A-ROD locus is already in close three dimensional proximity to the DKK1 locus and oligo capture of the nascent transcript of A-ROD may mediate a more indirect interaction with the DKK1 locus. Observing a decrease in A-ROD interaction at the DKK1 locus upon siRNA or ASO knockdown of A-ROD would confirm that recovery of the DKK1 locus is not due to three dimensional proximity with the A-ROD locus.

We thank the reviewer for this attentive comment, which gives us the opportunity to underline that we do not use the result of the ChIRP/CHART experiment to raise any argument whether it is the chromatin-associated or the nascent released form of A-ROD that interacts with the DKK1 locus, and in any case definitely not use it to discriminate and define the functionality of the two forms. In fact, both interactions could be captured by CHIRP/CHART, because of the crosslinking of the physical proximity and the design of the hybridization probes. Hence the ChIRP/CHART experiment is not employed to generate evidence whether it is the chromatin-associated or the released form of A-ROD that is the functional one. In the first place, the experiment was set up using hybridization probes that span the entire A-ROD transcript, almost all of them designed in exons, hence they do not provide the necessary discriminative power (when we give as input the whole A-ROD pre-mature transcript sequence, most of the optimal probes returned by the program are exonic, most probably because of sequence design restrictions in the intronic regions). Presuming that we capture both the chromatin-associated and released A-ROD, and that the siRNA treatment only impedes the released form, there could still be DNA signal co-eluted from the chromatin-associated fraction, captured as 'interacting' because of the crosslinked physical proximity. Would such a result merely assign the chromatin-associated A-ROD any functionality in DKK1 transcription regulation? Please note that we also co-elute A-ROD DNA, without meaning that A-ROD RNA transcript regulates transcription of each own locus. To improve the chance to use the ChIRP/ChART in such direction that the reviewer suggests, if possible, the siRNA knock-down would not be enough in such context, but additional refined experimental setup would be required necessitating the design of different sets of probes i.e. only exonic and only intronic. To optimize the design of purely intronic hybridization probes would be difficult because of sequence context restrictions (i.e. in general lower sequence complexity, repetitive elements etc.). In conclusion, we do not use the result of the ChIRP/CHART experiments to support functionality specifically for the released form of A-ROD. Nevertheless, to control for specificity we show that we cannot recover the DKK1 DNA locus when using biotinylated lacZ control oligos, hence the DNA recovery we obtain with the A-ROD antisense specific oligo probes is not merely due to chromosomal proximity captured by the crosslinking, but due to the RNA transcript capture.

4. To more-directly demonstrate a relationship between EBP1, delivered by "released" A-ROD, and DKK1 expression, EBP1 should be knocked down in both MCF-7 and MDA-MB-231 cells to test that only in the presence of A-ROD (MCF-7s) is EBP1 important for DKK1 expression. Without this, the fact that MDA-MB-231 cells express DKK1 in the absence of A-ROD call into question the context where A-ROD is important for DKK1 expression.

We thank the reviewer for this suggestion that we have followed using siRNAs against EBP1 in both MCF-7 and MDA-MB-231 cells. While the EBP1 knock-down efficiency is similar in both cell lines, DKK1 RNA level is impeded only in MCF-7 cells where A-ROD is expressed (Figure 9f).

5. Another means to increase the impact of this paper would be to further validate the computational pipeline by demonstrating a similar mechanism for an E2-activated lncRNA-mRNA pair. This would be particularly impactful because it is not actually clear what regulates A-ROD transcription itself.

We agree with the reviewer that it would be very interesting to find and confirm other candidates that would function like A-ROD (at their release from chromatin), while being E2-upregulated, but this will be the subject for substantial analysis for future studies. In this study we have not focused on E2-upregulated long ncRNA-target-gene pairs in the first place, for reasons that are mentioned in the text (lines 181-182, 188-191). Among other long ncRNAs, A-ROD is also E2-downregulated and we show that this repressive transcriptional response to E2 (*transcriptional* because of the use of GRO-seq data from Hah et al., 2013) precedes the response of DKK1 (Figure 3c). This in agreement with the experimental results of this manuscript supporting that A-ROD is required for

DKK1 transcriptional enhancement. Based on ERa ChIP-seq data (Welboren et al., 2009) we see that both A-ROD and DKK1 do not have ERa binding sites +/-20kb around their TSS, but how exactly the downregulation is exerted on A-ROD is not known, and would require substantial further experimental analysis that is out of the scope of this study.

6. Statistics are inconsistently presented. Most are shown in figure legends at beginning (i.e. Fig 2E – but these statistics are incorrectly mentioned in text line 168 and should include P-values for both comparisons or $P < \text{the lower P-value}$). Figures 4,5,6,7,8 have stars but no P-values shown anywhere. It is important to note statistical test used and P-values for each figure.

We thank the reviewer for this correction; we have added a description of p-values for each figure in the corresponding legend. 'Stars' notation follows the standard rule, 1 star $p\text{-value} < 0.05$, 2 stars < 0.01 , 3 star < 0.001 . We describe the statistical tests used in the Methods section.

7. Multiple editing issue exist in the manuscript. For some reason, every time a word should contain "ev", an "S" is in its place. Also, multiple notes from the authors to themselves are still in the manuscript. In most cases, these notes suggest changes that have not been made yet, but would improve the manuscript.

We apologize for this and all other mistakes in the manuscript and have carefully revised and rewritten the resubmitted manuscript.

Minor Points

1. Figure 1D needs better brackets to show clusters and a designation of "ChIA-PET" in the MaxScore column.

We have done this.

2. The PC analysis in Figure 2C could do with further description to highlight its value. Explanation of some of the trendlines that the data fall along would help, for example.

We have expanded the description of the PC results. In detail, in the caption of Figure 2c we explain what the plot represents, the meaning of direction and magnitude of the variable vectors, as well as the trendlines the data fall along. We have also modified the main text to guide the reader to a better interpretation of the PCA results and extended the Methods part giving more information on PCA.

3. The RT-qPCR analysis in Figure 4A is difficult to see in detail, and it is unclear which bars the stars of significance are supposed to mark. Some of these results could go in supplemental or be split into multiple figure panels.

We have removed the double and triple siRNA transfections from the figure panel to make it clearer and moved those data to Supplementary Fig. 4d.

4. It is unclear which GRO-seq data is used to filter expressed lncRNAs in MCF-7 cells.

We are citing the GRO-seq data in lines 91, 153, 172-173, 445; it is GSE43835 from Hah et al., Genome Res 2013.

5. The last two paragraphs of Results section should be moved to beginning of discussion.

Agreed and done.

Reviewer #2 (Remarks to the Author):

The major claim of this paper is that a lncRNA A-ROD needs to be released from chromatin in order to activate its target gene. This is indeed a novel claim and will contribute widely to our understanding of lncRNA function.

However, as written the manuscript is preliminary, in places rather ambiguous and I am not convinced that the data completely support the conclusions. This may be due to bad sentence construction as well as poor descriptions of methods and results which makes for difficult reading overall and making strong claims before the final bit of data is revealed. In order to communicate these data effectively to a wider community the manuscript requires substantial rewriting.

We thank the reviewer for the criticism and have substantially rewritten the manuscript and corrected mistakes throughout the text.

For me the exciting part of this work is that despite the close proximity of DKK1 and A-ROD brought about by chromatin conformation, the association is not influenced by expression of the lncRNA and that this confirms the work of others that have shown that chromatin conformation in many cases serves as a scaffold for regulatory interactions. (Intrachromosomal interactions have already in 2008 been described as scaffolds to recruit regulatory complexes Li et al 2008 MCB) I am also intrigued by the finding that the A-ROD transcript recruits EBP1 to the DKK1 promoter *in trans* to influence transcriptional elongation. These discoveries are what made me stick with reading this manuscript (many distracting typos and edits that have not been removed prior to submitting). Rather than the big focus on the "chromatin-release" the focus of the paper should centre on the EBP1 recruitment and the role an enhancer plays on transcript elongation as opposed to activation. The novelty here is how enhancing lncRNAs work (distinct from activating lncRNAs!)

We appreciate that the reviewer finds the manuscript exciting. We have to clarify here that we do not show that A-ROD acts *in trans*, but rather through a quasi-*cis* mechanism where release from chromatin is a functional prerequisite. A line of evidence supporting this is that the intronic ASOs do not dramatically reduce the steady-state nucleoplasmic levels of A-ROD within 24 h post-transfection, yet the DKK1 repressive transcriptional effect (assessed by BrU-labeled nascent RNA and Pol II P-Ser2 ChIP) is already evident. The mechanism we propose is that A-ROD is active (regulatory functional) upon release and within the pre-established chromosomal proximity (Figure 10). We find the chromatin release as a functional aspect of A-ROD to be the novel finding in our work, and have included additional data to elaborate both on the release of A-ROD from chromatin and the effects on DKK1 transcriptional elongation. For the detailed description of the new data, please see also response 2 to reviewer 1, where we describe additional experimental results of the revision, supporting our model.

Starting with the abstract - "chromatin-release" jumps out as a term which is not defined and could mean almost anything - the first thing I thought of was "releasing" chromatin looping associations between promoters and enhancers. It is only after the end of the 1st paragraph in the introduction that I began to understand that this work was examining whether the association of a lncRNA with chromatin was important for its role in regulating its target gene.

We thank the reviewer for this comment; as the first reviewer also suggested, in the revision we rather refer to dissociation of long ncRNAs from the chromatin-associated site of transcription or at "release from chromatin" (please see also response 1 to reviewer 1, and first response to the particular comments to the Results part below).

Introduction - enhancers, e-RNAs and activating lncRNAs are relevant, but this space could be better used to provide more background on the frequency of lncRNA associations/accumulation with chromatin at the site of transcription and some indication of what "chromatin-release of lncRNA" means: Does it infer that the lncRNA is associated with the chromatin post transcriptionally and then released, or does it mean the lncRNA is released into the nucleoplasm during transcriptional elongation? I prefer the terms chromatin associated lncRNA and nucleoplasmic lncRNA rather than released/disassociated lncRNAs

We have enriched the introduction with citations for long ncRNA chromatin-association as the reviewer requested (lines 42-47).

A long ncRNA, as any other Pol II produced transcript, cannot be released into the nucleoplasm before transcription elongation terminates and a mature 3' end is formed. We can detect some splicing of A-ROD in the steady-state chromatin-associated RNA-seq data (*de novo* transcript assembly prediction, Supplementary Fig. 5a-b), although most of the spliced form is nucleoplasmic enriched (Supplementary Fig. 8). That means that co-transcriptional processing at the A-ROD locus is efficient (well defined structure, splice sites and cleavage and polyadenylation site; please see also answer to comment 2 of reviewer 1) which most probably determines efficient nascent transcript release from the chromatin-associated template.

It is indeed a very interesting question for how long the long ncRNAs remain associated with the chromatin template post-transcriptionally, and based on the steady-state chromatin-associated and nucleoplasmic total RNA-seq data, we show that long ncRNAs transcribed from loci engaged in strong chromosomal interactions are less abundant in the chromatin-associated RNA fraction, suggesting a functional role for the chromatin-dissociation of those long ncRNAs.

Results

Expression and chromatin-release: This section is overly long should be rewritten and reordered - I am not sure whether the E2 induction of transcription is required in this section it disrupts the flow.

The K-means cluster analysis should be described in the order of 1. *De novo* transcript assembly of chromatin-associated (i.e the author's own data for lncRNAs identified after fractionating chromatin from nucleoplasm in MCF7 cells) and how it is filtered to produce a short list of 4,467 lncRNAs for further analysis 2. Summary of k-means clustering analysis (Fig 1A and D together with published data (GRO-seq data from Hah et al?) and ChiA PET PolII and promoter/enhancer histone signatures, followed by figs 1E, 2A, and 2B and C (2D and E can go into suppl data. The data so far only provide evidence that "cluster 6" lncRNAs with the highest promoter/enhancer and PolII Chia-PET score enrichments have higher nucleoplasm enrichment than chromatin association. I am struggling to follow the argument of chromatin-dissociation here despite the linear regression analysis - a lack of chromatin association does not mean "chromatin release" . Perhaps more careful analysis of the GRO-seq data may indicate whether lncRNAs at the point of transcription associates with chromatin or is nucleoplasmic - and would help with understanding whether the lncRNA is transiently associated or not. I certainly can't see how the data show a "relatively higher release" (line 148).

Regardless of the semantics here, Chia PET interactions with nucleoplasmic lncRNAs as opposed to chromatin associated lncRNAs does not support the statement that chromatin-release is important for functions of these lncRNAs line 169- 171.

We thank the reviewer for his suggestions to make our format more comprehensive. We took the advice and re-ordered the narrative (see new Figures 1, 2, 3). We have split the first results section and designated a separate section for differential chromatin association of long ncRNAs. Regarding the comment, "a lack of chromatin association does not mean chromatin release", we should clarify that we take the two conditions into account i.e. chromatin-associated and nucleoplasmic, and perform differential expression analysis using the chromatin-associated and nucleoplasmic RNA-seq data. This is not the first time that the term "release" is used in such context (e.g. Pandya-Jones et al., RNA 2013). When a long ncRNA is found mostly in the nucleoplasm it means that after its production it leaves chromatin. In fact, by performing differential expression analysis, we have extracted significantly chromatin-associated and significantly nucleoplasmic-enriched long ncRNAs (Supplementary Tables S2, S3). Long ncRNAs that are not post-transcriptionally efficiently released into the nucleoplasm are found as significantly chromatin-associated, like the recently studied *GNG12-AS1* (Stojic et al Nat Commun 2016); we also find it in our significant chromatin-associated long ncRNA dataset (Table S2). A-ROD on the other hand, like the rest of the entries of table S3, is a significantly nucleoplasmic enriched long ncRNA produced by a transcription unit which interacts in strong long-range chromosomal interaction with its target gene (DKK1).

<< I am struggling to follow the argument of chromatin-dissociation here despite the linear regression analysis - a lack of chromatin association does not mean "chromatin release" . Perhaps more careful analysis of the GRO-seq data may indicate whether lncRNAs at the point of transcription associates with chromatin or is nucleoplasmic - and would help with understanding whether the lncRNA is transiently associated or not. I certainly can't see how the data show a "relatively higher release" (line 148). >>

We would like to clarify here that the release is not measured by the GRO-seq data. GRO-seq data (taken from Hah et al., 2013) is a pure measurement of transcription activity, because it is global run-on sequencing of transcriptionally engaged RNA Pol II. The Pol II transcribed long ncRNAs, as any other Pol II transcribed molecule, is of course tethered to the chromatin-associated DNA template during transcription, and the whole transcriptional machinery functions on chromatin. In fact, what we measure and show using the GRO-seq data is that significantly chromatin-associated versus nucleoplasmic-enriched long ncRNAs do not show significant differences in their transcriptional states (Figure 2h, Supplementary Figure 2e). This is very important since it suggests that transcription activity *per se* does not determine chromatin dissociation (lines 156-163).

Line 173 -- this heading should read A-ROD and DKK are coordinately active in several cell lines. The conclusive sentence here line 195 is indicative of co-regulation not supportive of a tissue-specific activator.

True. This has been corrected in the revision where we stress that DKK1 expression is significantly higher in tissues where A-ROD is expressed, and the suggestion is followed by the phrase "as it was further analyzed". From next paragraph on, we describe the functional analysis supporting this statement, starting with the result that siRNA-mediated depletion of DKK1 does not affect A-ROD expression.

Line 197 - this heading implies that A-ROD activates DKK1 transcription but the data below only shows a conclusive effect transcriptional elongation (not activation).

We would like to clarify here that DKK1 transcriptional enhancement (or 'activation'; because it is not 'suppression' or 'repression') is at the level of productive transcription elongation and not at the transcription initiation step, as shown by Pol II P-Ser2 ChIP (Figure 6e-f) and TFIIB ChIP (Supplementary Figure 7c). To avoid perplexity we have adjusted the heading accordingly.

Line 211 "lncRNA target specificity" has nothing to do with the chromatin conformation data provided below. Further genomic information regarding enhancer and promoter elements as well as CTCF sites would be useful information for the reader to orientate the proximity interactions. In Fig 4C and D - why is the f2 site so variable between the MCF7 cells?

The target specificity seems to be determined by the chromosomal conformation and the interaction between A-ROD and DKK1 loci, so we find the headline appropriate for the chromatin conformation data. The differences at the f2 site in MCF7 cells come from independent experiments. As there are no evident differences at the surrounding regions we find this datapoint in agreement with no significant effect of the E2 treatment, despite the variable values.

Line 238 As the authors have just concluded based on the siRNA experiments in Fig 4D, that the lncRNA is not involved in structuring the chromatin conformation, chromatin release (or otherwise) can't be important here.

The reviewer refers to the line 236-238 in the first submitted manuscript, "Our analysis of chromatin-association of different groups of long ncRNAs suggests that chromatin-release is important for the function of long ncRNAs engaged in strong chromatin interactions (Fig 2A-B)". This sentence (now lines 248-250) is used to introduce the section "Regulation of DKK1 is not mediated by chromatin-tethered A-ROD" and refers to Figure 2 i.e. the first bioinformatics part of the paper and the results generated from analysis of the genome-wide data. Yes, based on the siRNA knock-down in the 3C experiment, neither transcription nor the long ncRNA transcript itself is involved in structuring chromosomal conformation. This has nothing to do with the functional importance of the dissociation of A-ROD from chromatin in transcription regulation of DKK1, and more broadly the implication of the chromatin-dissociation of long ncRNAs transcribed from loci engaged in strong long-range chromosomal interactions in the regulation of their target genes.

A piece of very important data is required? What is the actual Chromatin:Nucleoplasmic ration for A-Rod? Is the lack of effect of siRNA on chromatin (Fig 5A) due to low amounts of chromatin associated RNA and therefor beyond the resolution of detection?

We agree that this is important information. In Supplementary Figures 8 and 9 we have repeated the siRNA experiments and fractionation from new independent preparations and show the results prior to GAPDH normalization (also including addition control amplicons; Supplementary Figure 9), as the third reviewer also requested.

Based on the result of the differential expression analysis using the total chromatin-associated RNA-seq vs. nucleoplasmic RNA-seq, A-ROD is significantly nucleoplasmic enriched, this does not mean however that is not detected in the chromatin-associated fraction, where its expression is relatively high (UCSC genomic browser screenshot including our chromatin-associated RNA-seq data; Supplementary Fig. 4A). This, in combination with that the siRNA machinery does not efficiently work at chromatin may explain why we do not observe a robust knock-down on the steady-state chromatin-associated A-ROD. We do see it however by transfecting the RNase-H active intronic ASOs (Fig. 6A).

Why is the kd less efficient for A-ROD and DKK1 in Fig 5B compared to Fig 4A?

We thank the reviewer for this comment and have included a more detailed description of each of the figures. The main difference is that Fig. 4a shows whole-cell total RNA whereas Fig. 5b shows the nucleoplasmic fraction only. In fact, for the common 5'si and 3'si2 the respective differences in knock-down efficiencies are small and can be explained by the siRNA machinery being mostly active in the cytoplasm (please see also new Supplementary Fig. 9).

Rephrase lines 248 - 252 - there is no evidence (at this stage of the narrative) for whether the siRNAs are targeting elongation.

Thank you for this comment; we have deleted this sentence to erase any confusion for the readers.

Fig 5C does not show whether the nascent RNA was isolated from different cell fractions! In some cases siRNA located within 100 bp of the TSS may prevent transcription initiation-elongation (Stojic et al 2016 Nat. Comms). In the data presented here, this is not the case and that both the 3'and 5' siRNAs more effectively target A-ROD post transcriptionally or perhaps during elongation - NOTHING to do with the chromatin association.

Thank you for this comment. It was written in the text and figure legend, but now we have also included that the nascent RNA in Figure 5C is BrU-labeled chromatin-associated nascent RNA. In the case of the *GNG12-AS1* long ncRNA (Stojic et al Nat Commun 2016), that is a chromatin-associated long ncRNA, i.e. enriched at chromatin, and we note here that we also find it in our significantly chromatin-enriched long ncRNAs (Supplementary table S2). So, perhaps in that case, because the targeted long ncRNA does not leave the site of transcription, the siRNA machinery would 'have time' to target it while at chromatin.

Fig 6A - "data for DKK1"? Editing instructions still stuck in the ms? See also line 135 and 168! and 184!)

We apologize for these mistakes and have thoroughly corrected and rewritten the manuscript.

At Fig 6, is where the data is interesting! The ASO's targeted to the intronic regions reduces chromatin associated A-ROD but not nucleoplasmic A-ROD and still has an effect on nascent DKK1 expression (however, not at the activation stage, since the 1st DKK1 exon is transcribed! and also supported by the pausing index experiment and later by the Ser2 Pol II ChIP). Fig S6A is important here - However, if ASO is preventing the release of A-ROD from the chromatin, why then is there no difference in A-ROD levels in the nucleoplasmic fraction? Is this because there is such a small amount of Chromatin associated A-ROD that it doesn't impact upon the levels of nucleoplasmic transcript?

We would like to elaborate on this comment. The release is measured from the nascent (BrU-labeled) chromatin and nucleoplasmic fractions (now Supplementary Fig. 7B). The intronic ASO (targeting purely intronic A-ROD sequences; Supplementary Fig. 5A-B) does not have a major effect on the steady-state nucleoplasmic A-ROD, which is mostly in its spliced form, and we can detect only a small effect at later time-points post-transfection (>24h) (Fig. 6B). However, we

already see the repressive transcriptional effect on DKK1 within 24 h ASO post-transfection, that means that the steady-state nucleoplasmic form of A-ROD including spliced transcripts that have diffused away from the site of production, are not responsible for DKK1 transcriptional regulation, hence the mechanism is not *in trans*.

What effect would ASO's targeting exons or siRNAs targeting introns have on these experiments?

While the reviewer's question is in principle interesting, we haven't attempted either and rather focused on other experiments needing further elaboration to support the findings. siRNA targeting the introns would be inefficient since the siRNA machinery is the least efficient at chromatin; this is the reason why we followed the approach to target A-ROD intronic sequences with RNaseH-active ASOs. The A-ROD exonic ASOs would have a similar effect as the siRNA but would target both chromatin-associated and nucleoplasmic A-ROD, hence would lack any discriminative power.

I would have liked to have seen RNA Fish experiments with intronic and exonic probes to show nuclear and chromatin accumulation of A-ROD in addition to the CHIRP and CHART data in fig 8. Fig 8C is not particularly helpful, except to see that there were clearly more than 2 candidates.

We agree with the reviewer that FISH would have been interesting. We have attempted with a collaborator to do these experiments but could not obtain a satisfactory resolution.

If EBP1 is being recruited to the 1st exon of A-ROD genomic sequence (Fig 8E) then this does not match the model in Fig 8F. The conclusions would be strengthened and indeed validated by knock down experiments of EBP1 to see the effects on both transcripts as well as chromatin conformation.

We thank the reviewer for the suggestion and have included knock-down of EBP1 in both MCF-7 and MDA-MB-231 cells and show an MCF-7 (expressing A-ROD) specific effect on DKK1 RNA levels (Figure 9F, please see also response 4 to reviewer 1). In light of the negative results for dynamic effects on chromatin conformation we do not expect it to be informative to repeat the 3C experiment with EBP1 knock-down.

If A-ROD regulates DKK1 *in trans* as proposed, would we not expect ectopically expressed A-ROD to enhance DKK1 expression?

We do not propose that A-ROD regulates DKK1 *in trans*. We propose that A-ROD regulates DKK1 at its release from chromatin and within the pre-established proximity (Figure 10). Ectopically expressed A-ROD would most likely not have an effect on DKK1 expression.

The best part of the manuscript is the discussion which seems to have been written by a different author than the one who wrote the abstract, intro and results. However can the authors speculate how EBP1 binding to the promoter of DKK1 affects its transcriptional elongation.

We appreciate the reviewer's opinion about the discussion part. Regarding the EBP1 part, in this manuscript we have not reached the point to elucidate the precise molecular mechanism by which A-ROD enhances DKK1 transcription at the elongation step, which can be the subject for future studies. We show, that the regulatory function of A-ROD both in enhancing DKK1 transcription elongation and in recruiting a general transcription factor to the DKK1 promoter is mediated neither by the chromatin-associated form of A-ROD, nor *in trans*, but by the spliced form of A-ROD at its release from chromatin and within the pre-established chromosomal proximity.

Reviewer #3 (Remarks to the Author):

In this manuscript, Ntini et al. investigated the nuclear localization of long non-coding RNAs (lncRNAs) in MCF-7 breast adenocarcinoma cells and found that lncRNAs associated with strong chromatin interactions are less enriched in the chromatin fraction compared to other lncRNAs, which suggests a functional role in the release of lncRNAs from chromatin. The authors then focused on the A-ROD lncRNA and showed that the lncRNA does not mediate the chromatin interaction between the DKK1 gene and itself, but regulates transcription elongation of DKK1 after chromatin release instead.

On the whole, the manuscript feels more like a working draft rather than a complete piece of work. There are numerous comments strewn throughout, hinting at missing pieces of information. There is also a text substitution of "ev" into "S" throughout the entire manuscript. I would recommend the authors to use "lncRNA" as an abbreviation rather than "long ncRNA", especially since there is no mention of other non-coding RNA elements. The authors should also clearly note the datasets generated by themselves, and properly cite the public datasets used in the study.

We thank the reviewer for constructive criticism. We have thoroughly rewritten the text to correct the several mistakes as noted, and apologize for those.

We would like to use the term 'long ncRNAs' as more neutral, whereas the lncRNA abbreviation has been broadly used, and could mean, or even worse be mixed with, any of all the possible long ncRNA subtypes available, diverging even more among different classification systems (St Laurent et al., Trends Genet 2015).

We have emphasized the data we have generated, and cited the original literature or GEO numbers for all publicly available datasets generated by others.

1) The authors commented that there was a lag in the time response of lncRNAs compared to their target genes (Figures 1B and C), but the figures show the opposite trend. Furthermore, they compared Figures 1B and 1C while using linear and log Y-axis scales respectively. If both graphs are converted to linear Y-axis scales, there would arguably be a larger lag in the upregulated lncRNA-gene pairs instead of the smaller lag stated. Can the authors also provide the number of lncRNA-gene pairs considered in both graphs?

We thank the reviewer for this attentive comment. We have mixed the wording in the manuscript, of course it is the target genes that have a lag in the time response. We have corrected this in the revised version (lines 175-179). We have included the number of interacting pairs in the figure legend; it is 419 downregulated and 512 upregulated (with ChIA-PET interacting score ≥ 300). We agree with the reviewer that we should show both plots in the linear scale and so we have corrected that (now Fig. 3a-b). We see again that the long ncRNAs precede their target genes in the upregulated response and the lag is smaller compared to the downregulated pairs.

2) Can the authors show in detail how the chromatin-associated RNA Seq data was normalized to the nuclear poly-A+ RNA Seq data (Figure 2B)? I am uncertain about using RPKM ratio to compare the abundance of chromatin associated RNA and nuclear RNA. RPKM is an approximation of the RNA concentration within the sample (Wagner GP et al., Theory Biosci., 2012), but the original amount of RNA from both fractions would have differed quite significantly. As a reference, Werner et al. (Nat. Struct. Mol. Biol., 2017) compared the different fractions using spike in standards and absolute read counts.

The reviewer is right that RPKM values are not an ideal choice to compare normalized read counts between chromatin-associated RNA-seq data and nuclear polyA+ RNA-seq as we expect these two to differ substantially. In the absence of spike-ins or 'house-keeping' genes that could be used for corrections between the two fractions, the best we can do is to normalize for library size (as a proxy for the original RNA concentration). In addition, as we are comparing log ratios of scaled read counts (chromatin/nucleoplasm) across clusters, the normalization of the reads should not affect the relative comparison there. If (but actually we don't know) we are introducing a bias due to differences in RNA amounts in the fractions, we are introducing it consistently for every cluster. Therefore the relative comparison still holds and the conclusion from this plot does not change. We also performed the same analysis using the more robust normalization factor computed by DESeq2 (Love et al. 2014). This approach does not normalize for the initial library size or sequencing depth, neither for the gene length, but takes into account the distribution of the reads among the

input analyzed entries only (the long ncRNAs). By plotting the DESeq2 generated log₂ fold change of CHR-RNA-seq versus nuclear polyA+ RNA-seq we get a very similar distribution of the ratios among the different clusters (Supplementary Fig. 2a).

3) Can the authors mention in detail how A-ROD does not fit the characteristics of an enhancer-associated eRNA? (line 185-186)

We thank the reviewer for this question that gives us space for commenting. A-ROD is transcribed from a region that based on histone marks, DNA hypomethylation and ChIA-PET interaction can be characterized as an enhancer, so it is enhancer-transcribed, but does not function while enhancer-'associated' i.e. nascent spliced A-ROD has to leave the chromatin-associated site of transcription to exert function (please see also last paragraph of the discussion). In addition, according to previously established terminology, 'eRNAs' are described as short (<200nt), non-polyadenylated and short-lived transcripts arising mostly bidirectionally from enhancers (Kim et al., Nature 2010; Andersson et al., Nature 2014). We further address this point in the manuscript (lines 52-59, 196-202).

4) Can the authors show the expression of A-ROD and DKK1 (Figure 3B) in other breast cancer cell lines, such as T47D (ER+) and MDA-MB-231 (ER-), especially since MDA-MB-231 is used in the subsequent analyses?

We have included data for both MCF-7 cells and MDA-MB-231 in the revised manuscript. It is shown in the initial submission in the UCSC screenshots that MDA-MB-231 cells do not express A-ROD (Supplementary Fig. 4a). We now also show the rt-qPCR result (Supplementary Fig. 4e).

5) What is the proportion of A-ROD and DKK1 RNA in chromatin, nuclear and cytoplasmic fractions? Can the authors also show the siRNA knockdown of A-ROD and DKK1 in the chromatin, nuclear and cytoplasmic fractions, normalized to a single control? It may be that the low siRNA efficiency in the chromatin fraction is partly due to the low abundance of target RNA in that fraction (Hu X et al., Nucleic Acids Res., 2004) (Hong SW et al., Nucleic Acid Ther., 2014) (Lennox KA and Behlke MA, Nucleic Acids Res., 2016)

Please see Supplementary Fig. 8 for the proportion of A-ROD and DKK1 in the three fractions. In the revised manuscript we have repeated the siRNA experiments and fractionation from new and independent preparations, and show the fractionation result (cytoplasmic/nucleoplasmic/chromatin-associated) and the respective siRNA KD normalized to control (Supplementary Fig. 9).

We agree with the reviewer that due to the efficient release of the A-ROD transcript from the chromatin-associated site of transcription, in combination with the relatively low efficiency of the siRNA machinery at chromatin, we do not see efficient A-ROD knock down in that fraction, we do see it however by transfecting the intronic RNaseH-active ASOs (Figure 6a, Supplementary Fig. 7a).

6) I am not comfortable with the use of GAPDH as the reference gene for RT-qPCR in the different fractions of RNA. Since GAPDH is a protein coding gene, I would assume that its expression is not consistent between the different fractions. Can the RT-qPCR analysis be performed using one or more known chromatin-associated RNA as reference instead (with consistent expression in both chromatin and nuclear fractions)?

We would like to clarify here that we do not use GAPDH normalization to compare among different fractions. This would be wrong since GAPDH levels vary among different fractions (Supplementary Fig. 8). We only apply GAPDH normalization to correct for potential technical errors and slight differences in the RNA template amount used for the rt-qPCR when comparing among different biological replicates, control and knock-down conditions, and only within a given cellular fraction.

<< Can the RT-qPCR analysis be performed using one or more known chromatin-associated RNA as reference instead (with consistent expression in both chromatin and nuclear fractions)? >>

It is hard to find such a chromatin-associated RNA with consistent expression between the chromatin and the nucleoplasmic (nuclear?) fraction since per definition it would be enriched at chromatin. Nevertheless, we searched for transcripts that would have a similar abundance in the three fractions, or at least between chromatin and nucleoplasmic, when using the same amount (300ng) of RNA input in the rt-qPCR, that could potentially be used as "universal" reference genes,

but we could not find an ideal one (Supplementary Fig. 8). We also checked a previously reported gene that should have the same Ct value in the chromatin-associated and nucleoplasmic fraction (FNBP4; Werner and Ruthenburg, Cell Rep 2015) but in our hands in the rt-qPCR we find a higher expression level in the nucleoplasmic fraction (Supplementary Fig. 8). That could be because the cell line is different, since the fractionation controls behave properly (e.g. the GAPDH intron is significantly more abundant in the chromatin-associated fraction). Nevertheless, we repeated the siRNA experiments and fractionation from new and independent preparations, and in Supplementary Fig. 9 we show the result prior to any reference gene normalization, and solely normalized to the respective control condition within each fraction; this allows for comparison of the knock-down efficiency among the three different fractions, as the reviewer requested (please see also response to comment 6).

7) Why did the authors decide not to show the methylation status of the A-ROD promoter in MCF-7 and MDA-MB-231 cells? (line 222)

We now show the methylation status of the A-ROD promoter in MCF-7 from our bisulfite sequencing data, and from the ENCODE methylation data, as well as the sequencing electropherogram from the locus-specific PCR of bisulfite-converted DNA in MCF-7 and MDA-MB-231, as the reviewer requested (Supplementary Fig. 4f).

8) Which siRNAs were used in Figure 4D? Will it be possible to show the knockdown effects on MDA-MB-231 as well?

That was siRNA 3'si2; we have included this information in the figure legend and figure. We haven't done the 3C in the presence of A-ROD siRNA in MDA-MB-231 because A-ROD is not expressed in this cell line (please see also response to comment 4).

Reviewers' comments:

Reviewer #1 (Remarks to the Author):

The revised manuscript by Ntini et al. is significantly improved, both in additional experiments and text revisions. The authors have incorporated experiments disrupting the biogenesis of the A-ROD lncRNA, both blocking cleavage/polyadenylation and splicing, and this leads to a negative effect on the elongation of the DKK1 gene. They also provide evidence that A-ROD presence correlates with the requirement for EBP1 to regulate DKK1. At this point, the experiments are fleshed out to the point where the model is supported sufficiently. The actual model is intriguing and is a valuable addition to the field. However, a thorough editing of the main figures of the manuscript, with a few experiments added for consistency, is required to more-clearly provide the key information to support the final model. Below are a series of suggestions that should be considered to revise this manuscript adequately.

Suggestions:

1. If possible, some figures should be combined and some data moved to supplemental to focus on the main points of the paper and increase clarity. For example, Figures 1 and 2 may work best as a streamlined single figure.
2. Similar experiments in the main figures should be consistently performed, wherever possible. For example, in previous figures, the chromatin and nucleoplasmic fractions are profiled when using siRNAs; however in Figure 7 only BrU-CHR RNA levels are determined. The authors frequently use each of these conditions to make the point about chromatin release, therefore this consistency should be maintained and the results commented on.
3. Paring down the number of siRNA/ASO sets and morpholino concentrations used and the number of amplicons profiled in Figs. 6,7,8 to emphasize the main findings, while maintaining sufficient controls would greatly increase clarity of interpretation.
4. Further clarification is needed in the text to explain the following regarding the CPA/PAS blocker and morpholinos experiments:
 - a. What accounts for the differences in morpholino versus CPA/PAS blockers? Are there cryptic cleavage sites that are used that lead to some release of A-ROD when CPA/PAS are blocked, but morpholinos are more-effective at altering mature A-ROD levels?
 - b. Also, why do the morpholinos only affect Pol II CTD Ser2-Phos and not DKK1 last exon RNA levels?
5. Figure 7 and 8 could easily be incorporated into a single figure, with some effort put into editing to put only the most-important information in this main figure.

Reviewer #2 (Remarks to the Author):

Compared to the initial submission I am indeed impressed by the revised manuscript.

All of my comments have been addressed. It reads well (apart from some unnecessarily long sentences) and is markedly more convincing than the initial submission.

I suggest another careful reading of all the figure legends to correct minor typos.

I particularly like the model figure.

The abstract still doesn't really do the study justice.

Reviewer #3 (Remarks to the Author):

My comments have been adequately addressed and I feel the manuscript is ready for publication.

Response to reviewers:

Reviewer #1 (Remarks to the Author):

The revised manuscript by Ntini et al. is significantly improved, both in additional experiments and text revisions. The authors have incorporated experiments disrupting the biogenesis of the A-ROD lncRNA, both blocking cleavage/polyadenylation and splicing, and this leads to a negative effect on the elongation of the DKK1 gene. They also provide evidence that A-ROD presence correlates with the requirement for EBP1 to regulate DKK1. At this point, the experiments are fleshed out to the point where the model is supported sufficiently. The actual model is intriguing and is a valuable addition to the field. However, a thorough editing of the main figures of the manuscript, with a few experiments added for consistency, is required to more-clearly provide the key information to support the final model. Below are a series of suggestions that should be considered to revise this manuscript adequately.

We thank the reviewer for helpful suggestions during the first round of review that have improved the manuscript and supported our model further. We have carefully considered all of the suggestions from second round of review and have implemented some of these. We are grateful to the reviewer that s/he puts effort into improving our paper, both at the scientific and presentation level.

Suggestions:

1. If possible, some figures should be combined and some data moved to supplemental to focus on the main points of the paper and increase clarity. For example, Figures 1 and 2 may work best as a streamlined single figure.

We agree with the reviewer and have merged the first two figures as suggested.

2. Similar experiments in the main figures should be consistently performed, wherever possible. For example, in previous figures, the chromatin and nucleoplasmic fractions are profiled when using siRNAs; however in Figure 7 only BrU-CHR RNA levels are determined. The authors frequently use each of these conditions to make the point about chromatin release, therefore this consistency should be maintained and the results commented on.

To assess targeting specificity and compare the effects of the two different knock-down approaches, siRNAs and ASOs, we use both the chromatin and nucleoplasmic fractions. This is necessary in order to firmly support our conclusions on release from chromatin based on which we draw our model, *e.g.* lines 305-311 in the marked manuscript (or lines 300-306 in the unmarked version of the manuscript uploaded as related manuscript file). To assess transcription (and transcription read-through) we use only the BrU-labeled chromatin associated nascent RNA (revised Figures 4c, 5c, 6b), supported as well by P-Ser2 PolII ChIP. This is done on purpose, as transcription takes place at chromatin and is not an inconsistency in the manuscript. The data were presented this way also in the initial submission (for revised Figures 4c and 5c; Figure 6b has been added in revision).

3. Paring down the number of siRNA/ASO sets and morpholino concentrations used and the number of amplicons profiled in Figs. 6,7,8 to emphasize the main findings, while maintaining sufficient controls would greatly increase clarity of interpretation.

We thank the reviewer for this suggestion. To maintain a sufficient level of controls we prefer to keep the figures as they are. We find it necessary to provide results from several biological replicates; assaying several different siRNA and ASO sequences; and include several different amplicons in order to adequately support our conclusions. We find that these data would be misplaced in the Supplementary information. Following the reviewer's suggestion in the first round of revision, we have already moved some otherwise relevant data to the Supplementary information aiming at a balance between the required proof from biological repetition outcome and minimized perplexity.

4. Further clarification is needed in the text to explain the following regarding the CPA/PAS blocker and morpholinos experiments:

a. What accounts for the differences in morpholino versus CPA/PAS blockers? Are there cryptic

cleavage sites that are used that lead to some release of A-ROD when CPA/PAS are blocked, but morpholinos are more-effective at altering mature A-ROD levels?

We thank the reviewer for this comment. In fact, there is only considerably slight difference regarding the efficiency of the two approaches on the extent of the caused transcriptional downregulation effect, as this is captured only by the amplicon +300 of DKK1. We should note here that these are two different experimental approaches of distinct chemistry conducted for different treatment periods (24 h 2'OMePS treatment to block PAS/CPA versus 36 h MO treatment to inhibit splicing). Hence, the difference observed only for one amplicon of DKK1 does not allow us to draw strong statements regarding differences in the efficiencies. The important outcome is that both approaches consequently induce a repressive transcriptional downregulation effect on DKK1. Nevertheless, we agree with the reviewer that there could be some cryptic PAS/CPA sites that could become utilized when the major PAS/CPA is blocked.

b. Also, why do the morpholinos only affect Pol II CTD Ser2-Phos and not DKK1 last exon RNA levels?

We thank the reviewer for this comment. The splice-inhibiting morpholinos efficiently reduce the nascent spliced released form of A-ROD at 36 h treatment, consequently causing a repressive effect on DKK1 transcription, measured by P-Ser2 Pol II ChIP. However, this is not effectively reflected neither at the steady-state chromatin-associated nor at the steady-state nucleoplasmic RNA levels of DKK1 at this time point post-transfection.

5. Figure 7 and 8 could easily be incorporated into a single figure, with some effort put into editing to put only the most-important information in this main figure.

We have merged the two figures into new main Figure 6, as the reviewer suggested.

Reviewer #2 (Remarks to the Author):

Compared to the initial submission I am indeed impressed by the revised manuscript.

All of my comments have been addressed. It reads well (apart from some unnecessarily long sentences) and is markedly more convincing than the initial submission.

I suggest another careful reading of all the figure legends to correct minor typos.

We have corrected minor typos in figure legends. We have had a colleague with English as his native language to read through the manuscript and suggest improvements. All changes to the text are highlighted in the revised manuscript version.

I particularly like the model figure.

Thank you.

The abstract still doesn't really do the study justice.

We have improved the abstract to capture the major conclusions of the paper and highlight the importance of the proposed model of regulation.

Reviewer #3 (Remarks to the Author):

My comments have been adequately addressed and I feel the manuscript is ready for publication.

We thank the reviewer.

REVIEWERS' COMMENTS:

Reviewer #1 (Remarks to the Author):

The manuscript is improved in clarity and structure. The authors have addressed all of my comments sufficiently.

REVIEWERS' COMMENTS:

Reviewer #1 (Remarks to the Author):

The manuscript is improved in clarity and structure. The authors have addressed all of my comments sufficiently.

We thank the reviewer for his/her helpful comments throughout the review process that has helped to improve the manuscript.